# Can DBNNs Robust to Environmental Noise for Resource-constrained Scenarios?

**Wendong Zheng** [1]   **Junyang Chen** [* 2]   **Husheng Guo** [1]   **Wenjian Wang** [* 1]

## Abstract

Recently, the potential of lightweight models for resource-constrained scenarios has garnered significant attention, particularly in safety-critical tasks such as bio-electrical signal classification and B-ultrasound-assisted diagnostic. These tasks are frequently affected by environmental noise due to patient movement artifacts and inherent device noise, which pose significant challenges for lightweight models (e.g., deep binary neural networks (DBNNs)) to perform robust inference. A pertinent question arises: can a well-trained DBNN effectively resist environmental noise during inference? In this study, we find that the DBNN's robustness vulnerability comes from the binary weights and scaling factors Drawing upon theoretical insights, we propose L1-infinite norm constraints for binary weights and scaling factors, which yield a tighter upper bound compared to existing state-of-the-art (SOTA) methods. Finally, visualization studies show that our approach introduces minimal noise perturbations at the periphery of the feature maps. Our approach outperforms the SOTA method, as validated by several experiments conducted on the bio-electrical and image classification datasets. We hope our findings can raise awareness among researchers about the environmental noise robustness of DBNNs.

## 1. Introduction

To enable safety-critical tasks (e.g., traffic flow control, bio-electrical monitoring, etc.) (Hsiao et al., 2022; Qiu et al., 2025; Roeder, 2022) on edge devices with limited computational resources, researchers often consider deploying lightweight models. This implies that lightweight models are typically trained on GPU servers, after which the well-trained models are deployed to edge devices for performing downstream inference tasks. In particular, edge devices are typically in environments with diverse forms of environmental noise (Yayla & Chen, 2022). If environmental noise causes the lightweight model to output wrong results, it can have serious consequences for safety-critical tasks. For examples, the smart bracelet worn by the patient introduces environmental noise perturbations, which is compounded by the patient's pain and other physiological conditions (Roeder, 2022). Thus, we aim to support robust decision-making of lightweight models under environmental noise perturbations, thereby assisting decision-makers.

**Limitations:** On Resource Limited Scenarios, the robustness of lightweight models with sparse property is compromised in environmental noise perturbations (Cai et al., 2022). Empirically, several researches are focused on adversarial learning schemes (Miyato et al., 2018; Gouk et al., 2021) using spectral norm constraints to improve the robustness of full-precision models. In theory, recent approaches extensively employ relaxation strategies based on the Lipschitz continuity theorem to establish provable bounds for full-precision convolutional neural networks (CNNs) against adversarial attack (Gowal et al., 2018; Balunovic & Vechev, 2020; Zhang et al., 2020; 2021). However, this kind of method mainly has the following three limitations. (1) These relaxation-based methods (Gowal et al., 2018; Balunovic & Vechev, 2020) improve the robustness of full-precision models when deployed on GPU servers, yet they introduce exponential computational overhead. Thus, the aforementioned approaches (Gowal et al., 2018; Balunovic & Vechev, 2020) present significant challenges for deployment in resource-constrained environments. (2) The above method (Gowal et al., 2018; Balunovic & Vechev, 2020; Zhang et al., 2020; 2021) exhibit effectiveness only against specific types of noise perturbations, owing to their training with specific adversarial samples. However, the nature of environmental noise that poses challenges for safety-critical tasks is difficult to predict. For instance, a patient's position change due to discomfort can introduce random noise into the electrodes collecting bio-electrical signals (Roeder, 2022). (3) The

---

*Co-corresponding author   [1]The School of Computer and Information Technology, Shanxi University, Taiyuan, China [2]The College of Computer Science and Software Engineering, Shenzhen University, Shenzhen, China. Correspondence to: Junyang Chen <junyangchen@szu.edu.cn>, Wenjian Wang <wjwang@sxu.edu.cn>.

*Proceedings of the 42nd International Conference on Machine Learning*, Vancouver, Canada. PMLR 267, 2025. Copyright 2025 by the author(s).

above approaches (Gowal et al., 2018; Balunovic & Vechev, 2020; Zhang et al., 2020; 2021) focus on the shallow CNNs. Due to the structural disparities between shallow and deep model, it remains challenging to determine whether existing methods can be directly applied to analyze the robustness of deep backbone (e.g., ResNet34).

**Challenges:** The utilization of binary neural networks (BNNs) as an extreme approach for quantifying weights and activations has yielded remarkable success in several machine learning tasks (Rastegari et al., 2016; Zhou et al., 2016; Liu et al., 2018; Bulat & Tzimiropoulos, 2019; Liu et al., 2020; Qin et al., 2020; Tu et al., 2022; Yuan & Agaian, 2023; Qin et al., 2023). The study (Cai et al., 2022) shows that the quality of inference from lightweight models are observed to decrease on edge devices with environmental noise. Especially in some medical assisted diagnosis scenarios (Roeder, 2022), noise interference has a serious negative impact on decision making. However, the existing approaches focus on narrowing the performance gap between the full-precision and BNNs, without delving into the robustness of BNNs against environmental noise perturbations during inference. Compared to full-precision models, the LCR method (Shang et al., 2022) has investigated the insufficient robustness of BNNs. However, it still encountered two challenge issues. (1) The study (Shang et al., 2022) introduce a large extra overhead to BNNs, which is difficult to meet the requirements of resource limited scenarios. In particular, the approximation operation using several retention matrices result in a 20% increase in training overhead. (2) The study (Shang et al., 2022) lacks a quantitative formal analysis of noise perturbation bounds to demonstrate the robustness of DBNNs.

**Motivation & Contribution:** To address the above-mentioned challenge issues, we aim to devise an upper bound via $L_{1,\infty}$-norm that can analyze the robustness of DBNNs under environmental noise perturbations. Furthermore, the analysis also aim to reveal that the vulnerability in DBNNs model robustness originates from the scaling factor associated with binary operations. In contrast to the recent study (Shang et al., 2022) by using an approximate spectral norm constraint strategy, our proposed $L_{1,\infty}$-norm constraint only involves the column and row of matrix and the maximum computing operations, thus avoiding additional training costs like in the study (Shang et al., 2022). In order to better target the characteristics (i.e., deep backbones and binarization operations) of DBNNs under environmental noise (i.e., random and high SNR), our theoretical analysis framework by using $L_{1,\infty}$-norm offer superior applicability to DBNNs in mitigating the negative impact of environmental noise. Meanwhile, we consider the characteristics of the DBNNs (e.g., scaling factors and binarization operations) to provide a tightness upper bound under environmental noise perturbations. Finally, to address the potential risk

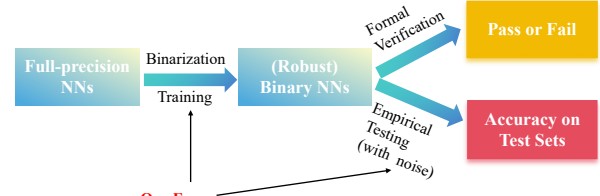

*Figure 1.* Workflow for model binarization and robustness evaluation.

of our auxiliary robustness loss function (i.e., requiring the maximization of a non-concave function over a norm ball), we introduce a constraint coefficient associated with scaling factors of DBNNs to balance the classification and auxiliary robustness loss functions (See **Appendix** for detailed experiments). Our main contributions are summarized as follows:

- In theory, we find that the vulnerability of DBNNs to environmental noise comes from scaling factors and binary weights. Then, we formally derive an noise perturbations upper bound for DBNNs in a closed-form analytical manner. We provide a tighter upper bound compared to the SOTA method under same noise conditions.

- We propose a novel automated and robust training framework to enhance the environmental noise robustness of various DBNNs by using the $L_{1,\infty}$-norm constraint.

- Our visual case finds the detrimental impact of environmental noise on the feature maps of DBNNs.

Experiments show that our approach enhance the robustness of DBNN-based models on five classification tasks, with maximum improvements of 4.8% and 5.4% on the CIFAR-100, Brain tumor MRI datasets, respectively.

**Discussion summary:** Our work aims to provide an effective training framework for enhancing the robustness of DBNNs against environmental noise by incorporating an $L_{1,\infty}$ norm constraint training, hence it is orthogonal to the topic of formal verification for verifying the correctness of the resulting quantized/binary NN (In Fig.1). Please see **Appendix** for details.

## 2. Related Works

### 2.1. Binary Neural Networks

The current radical compression strategy involves reducing the learnable parameters of bit-width from 64 to 1-bit (Yuan & Agaian, 2023; Qin et al., 2023). Several researchers (Rastegari et al., 2016; Zhou et al., 2016; Liu et al., 2020;

2018; Qin et al., 2020; Tu et al., 2022) foremost compress weights $w_B \in R$ and activations $a_B \in R$ (detailed in survey work (Yuan & Agaian, 2023)) to 1-bit in each convolution block. In comparison to conventional compression method (i.e., pruning), binarization exhibits superior computation operators as it exclusively targets the model parameters (Rastegari et al., 2016). However, the binarization brings a scarcity of feature representation, leading to a significant degradation in accuracy. Now, the study (Qin et al., 2023) proposes a method that leverages information gain to mitigate these representation losses and thus enhance the performance of BNNs. However, these methods do not consider whether BNNs directly used for inference is resistant to environmental noise.

## 2.2. Robust Neural Networks using Lipschitz Continuity

Recently, the study (Gouk et al., 2021) proposes an effective approach to enhance the performance of feed-forward neural networks by computing an upper bound on the Lipschitz constant for multiple $p$-norms. Especially in the area of image processing, the study (Yoshida & Miyato, 2017) proposes spectral normalization to constrain the Lipschitz constant of the discriminator by optimizing a adversarial network. Meanwhile, several studies (Li et al., 2019; Singla & Feizi, 2021) propose strategies to effectively reduce the Lipschitz constant of CNNs by leveraging $L_2$-norm. Then, the study (Zhang et al., 2021) further proposes a novel network architecture, termed $L_\infty$-dist network structure, by leveraging inherently 1-Lipschitz functions as neuron units. The core module of the study (Miyato et al., 2018) is noteworthy for its utilization of spectral norm constraints based on singular values, which are directly applied to the weights of each convolutional layer. However, the extreme sparsity of binarized weights pose an immediate computational challenge by directly via spectral norm constraints (Shang et al., 2022). To resolve this issue, a study (Shang et al., 2022) aimed at enhancing the robustness by incorporating Lipschitz continuity constant as a regularization term. To address the extreme sparsity of binary weights, this work (Shang et al., 2022) proposes employing retention matrices to approximate spectral norms of weights. However, the effectiveness of the LCR (Shang et al., 2022) heavily relies on the strong mutual linear independence assumption of binary activations.

## 3. Background

**Robustness Definition of BNNs:** Following the theoretical study (Narodytska, 2018), we adopt the $L_\infty$-norm as a metric to quantify distance. From a theoretical perspective, global robustness represents an exceedingly stringent property. Considering real-world scenarios, our research is dedicated to enhancing the local robustness of DBNNs,

which can be defined as follows:

**Definition 3.1.** ($L_\infty$-robustness) Let $F(x)$ represent the output of neural network $F$ on input signals $x$ and $\ell_x = L(x)$ be the ground truth of $x$. If the $F$ is locally robust, there is not exist perturbed $\gamma$ that makes $||\gamma||_\infty$ holds on. e.g., we have $F(x + \gamma) \neq \ell_x$.

Secondly, we outline the prevalent characteristics of BNN-based models (Rastegari et al., 2016; Liu et al., 2018; 2020; Qin et al., 2020; Yuan & Agaian, 2023; Qin et al., 2023), which involve the conversion of full-precison weights ($W_F$) and/or activations ($I_F$) into 1-bit precision. Here, the standard definition (Qin et al., 2023; Liu et al., 2018; Qin et al., 2020) of the binarization operation is provided as follows:

$$Q(W_F) = \alpha * W_b, \ Q(I_F) = \beta * I_b, \tag{1}$$

where $W_b$ and $I_b$ are denote the binary weight and binary activation matrices, respectively. Here, the $\alpha = \frac{1}{n}\|W_F\|_1$ and $\beta = \frac{1}{n}\|I_F\|_1$ are represent two scaling factors for full-precision weight ($W_F$) and activation ($I_F$). Following the survey (Yuan & Agaian, 2023), we present the description of the $\text{Sign}(\cdot)$ function:

$$\text{Sign}(x) = \begin{cases} +1, & \text{if } x \geq 0 \\ -1, & \text{otherwise} \end{cases}. \tag{2}$$

Furthermore, we give a general computation process of DBNNs in Def.3.2.

**Definition 3.2.** The BNNs typically employs binary $\{-1, 1\}$ weights and activations components to achieve light-weight modeling. Then, we can write such network thought $L-1$ modules (i.e., $(\alpha*I_b^l)\otimes(\beta*W_b^l), l \in [1, L-1]$, where $\otimes$ denotes the convolution operation) with weights and activations and $L$-th layer (i.e., output module $O(\cdot)$). Hence, we can deduce the iterative form of BNNs with input $x$ as follows:

$$\text{BNNs}(x) = O((\alpha I_b^{L-1}) \otimes (\beta W_b^{L-1}) \cdots (\alpha I_b^1) \otimes (\beta W_b^1(x))). \tag{3}$$

**Notations of Matrix and Perturbed Robustness:** We can represent matrices by uppercase letter, such as $W$. For a weight matrix $W$, we can write its $i^{th}$ row and $j^{th}$ column as the entry $W_{i,j} \in \mathcal{R}^{m \times n}$. The $L_\alpha$ norm of a matrix can be abbreviated as $\|W\|_\alpha$. The $L_{1,\infty}$-norm will be used in the subsequent theoretical analysis of robustness. According to the definition of the matrix norm, we can get the formal definition for $L_{1,\infty}$-norm $\|A\|_{1,\infty} = \max_{\vec{x} \neq \vec{0}} \frac{\|A\vec{x}\|_1}{\|\vec{x}\|_\infty}$, where $\vec{x}$ denotes a vector and $A$ denotes a matrix. Recently, the study (Mao et al., 2023) has focus on the $L_\infty$ robustness of perturbed inputs as follows:

$$B_x^\infty(\epsilon) \stackrel{\text{def}}{=} \{x' | \|x' - x\|_\infty \leq \epsilon_x\}, \tag{4}$$

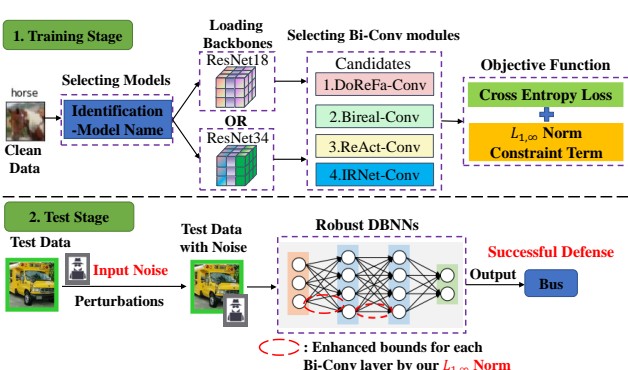

Figure 2. Our automatic robust training framework.

where $x'$ denotes a series of perturbed inputs and $x$ denotes a series of clean inputs, the $\epsilon_x$ denotes a input noise perturbation radius and $B_x^\infty(\epsilon)$ represents a $L_\infty$ norm ball of the noise perturbation radius $\epsilon_x$.

## 4. Method

**Overview:** In this section, we propose a novel robust training method that utilizes a targeted objective function with a $L_{1,\infty}$-norm constraint term on binary weights to enhance the robustness of DBNNs against environmental noise perturbations. In contrast to the work (Shang et al., 2022), our proposed method effectively mitigates the excessive training cost resulting from the indirect preservation of the matrix. To analyze more suitable constraints, we establish a quantitative relationship between our perturbation bounds and the bounds of study (Shang et al., 2022). To facilitate the application of our approach for enhancing the robustness of various BNN structures, we propose an automatic training framework (See the last paragraph of section 4.3 for details) as shown in Fig.2.

### 4.1. Formal Analysis on the Robustness of DBNNs

**Main Analysis Process:** (1) We present a feed-forward and full-precision neural network under general noise perturbations as an example in Fig.3. Here, the weights $W_{1,1}$, $W_{1,2}$ should be annotations on the edges. (2) Taking a simple 3-layer full-precision model as an illustrative example, we present an analysis of the disparity in learnable weights between the outcomes of the two categories before and after introducing input noise perturbations. (3) Based on the above theoretical findings and the inherent characteristics of the DBNNs (e.g., scale factors $\alpha, \beta$ and binary weights $W_b$), we can deduce the upper bound of $L$-layer DBNNs model under noise perturbations.

**Step1: The Process of Full Precision Models with Noise.** In Fig.3, our example illustrates the variations in each weight value under noise perturbations, thereby offering a valuable contribution to the subsequent analysis of the

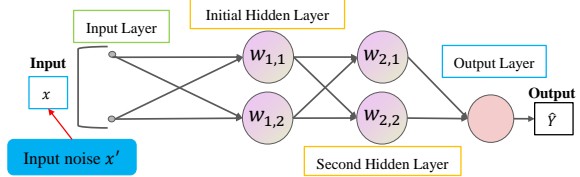

Figure 3. The example of a full-precision neural network perturbed by noise.

upper bound of DBNNs. In Fig.3, the noise is applied to the image signal $x$. Subsequently, we demonstrate the disparity between the output of the DBNNs under clean and noise scenarios. Specifically, in Eqn.5, $\hat{Y}$ represents the perturbed output resulting from two examples with noise (i.e., $\hat{x}_1$ and $\hat{x}_2$). Here, $|\hat{Y} - Y|$ denotes the absolute difference between the disturbed and clean outputs. Then, ReLU($\cdot$) denotes the ReLU activation function. In addition, $W_{1,2}$ denotes to the second weight value in the initial hidden layer.

$$|\hat{Y} - Y| = |W_O \text{ReLU}\{\text{ReLU}(\hat{x}_1 W_{1,1})W_{2,1}$$
$$+ \text{ReLU}(\hat{x}_1 W_{1,2})W_{2,2} + \text{ReLU}(\hat{x}_2 W_{1,1})W_{2,1}+$$
$$\text{ReLU}(\hat{x}_2 W_{1,2})W_{2,2}\} - W_O \text{ReLU}\{ \qquad (5)$$
$$\text{ReLU}(x_1 W_{1,1})W_{2,1} + \text{ReLU}(x_1 W_{1,2})W_{2,2}$$
$$+ \text{ReLU}(x_2 W_{1,1})W_{2,1} + \text{ReLU}(x_2 W_{1,2})W_{2,2}\}|.$$

**Step2: Formal Analysis for Full Precision Models under Noise Perturbations.** For the first hidden layer of a full precision model, we quantify the discrepancy through applied noise perturbations (also referred to as the model's decision margin) with $L_\infty$ perturbed inputs based on Eqn.4-Eqn.5 are as follows:

$$e^1 \stackrel{\text{def}}{=} \sum_{i=1}^{1} \sum_{j=1}^{2} |W_{i,j} \cdot (\hat{x}_j - x_j)| \leq \sum_{i=1}^{1} \sum_{j=1}^{2} |W_{i,j}| \cdot |\hat{x}_j - x_j|$$

$$\leq \epsilon_x \max_i \sum_{i=1}^{1} \sum_{j=1}^{2} |W_{i,j}| = \epsilon_x \cdot \|W^1\|_\infty,$$
$$(6)$$

where $e^1$ denotes the difference between clean and applied noise at the initial hidden layer. The weight value of the $i^{th}$ row and $j^{th}$ column in a matrix is denoted as $W_{i,j}$, while the initial hidden layer of all weights is represented by $W^1$. Then, $\epsilon_x$ represents the perturbation value for inputs. It is imperative to maximize this disparity to effectively constrain the model and enable correct decision against noise perturbations. Hence, we provide an expanded upper bound on the right side of the definition and then employ the $L_\infty$ norm to calculate its maximum value.

Thereafter, we present the perturbation bounds of second hidden layer from the preceding layer with respect to input $\epsilon_x$ as follows:

$$e^2 \leq \text{ReLU}(\underbrace{\epsilon_x \cdot \|W^1\|_\infty}_{e^1}) \cdot \|W^2\|_\infty, \qquad (7)$$

*Table 1.* The distinctions between our study and previous studies. Here, $*$ denotes the additional training cost.

| Methods | Constraints | Types | Deep Structures | Assumptions | Additional Cost |
|---|---|---|---|---|---|
| ELC (2021) (Gouk et al., 2021) | $L_p$-norm | DNNs | VGG19 | N/A | Not |
| SN (2018) (Miyato et al., 2018) | $L_2$-norm | DNNs | ResNet18 | N/A | Not |
| LCR (2022) (Shang et al., 2022) | $L_2$-norm | BNNs | ResNet20 | stringent | **+20% cost**$^*$ |
| Our | $L_{1,\infty}$-**norm** | DBNNs | **ResNet34** | N/A | Not |

where $\|W^2\|_\infty$ denotes the difference for the second hidden layer. Finally, we can provide the perturbation bounds of output layer based on Eqn.7 as follows:

$$e^3 \le \text{ReLU}(\text{ReLU}(\epsilon_x \cdot \|W^1\|_\infty) \cdot \|W^2\|_\infty) \cdot \|W_O^{c1} - W_O^{c2}\|_1 \cdot \tag{8}$$

According to the study (Gouk et al., 2021), we employ the $L_1$ norm to quantify the discrepancy in weights between a perturbed output result and an unperturbed output result as $\|W_O^{c1} - W_O^{c2}\|_1$ in Eqn.8.

**Step3: Formal Analysis for Environmental Noise Bounds of $L$-layer DBNNs.** Based on the Def.3.2, we focus on the DBNNs using the classical ResNet-based backbone. Then, the formal description perspective for DBNNs only requires the conversion of activation and weights from full-precision models in Eqn.7 to binarized form.

**The Difference between Full-precision models and DBNNs:** The primary differents between the computational processes of full-precision model (i.e., $\text{ReLU}(W)$) and BNNs lie in the binarization of activations (i.e., $\beta I_b$) and weights (i.e., $\alpha W_b$) with scaling factors. Hence, we can illustrate a BNNs example of final bounds via the Def.3.2 and Eqn.8 as follows:

$$e_b^3 \le \{\beta I_b \otimes (\epsilon_x \alpha \|W_b^1\|_\infty) \beta I_b \otimes (\alpha \|W_b^2\|_\infty)\} \Delta W_b^o, \tag{9}$$

where $W_b^2$ denotes the all binary weights of the second binary convolution layer, and then $\Delta W_b^o = \|W_{b;c1}^o - W_{b;c2}^o\|_1$ denotes the difference of binary weights corresponding to different classification results at output layer by utilizing $L_1$ norm. Here, $\alpha$ and $\beta$ denote scaling factors. Thus, we can obtain the upper bound of the DBNNs under environmental noise in Def.4.1.

**Definition 4.1.** Suppose a standard classification task contains $P$ categories, such as $C_i, i \in [1, \cdots, P]$. Suppose the output of the DBNNs is $C_1$ instead of the expected result $C_2$, we can define the discrepancy between different classification results using $L$-layer DBNNs under environmental noise perturbations as follows:

$$F_{W_b}^{C_1}(\hat{x}) - F_{W_b}^{C_2}(x) \overset{\text{def}}{=} (\frac{1}{n} \|W_F\|_1 \cdot W_{b;C_1}^L \cdot \hat{h}_B^{L-1})$$
$$- (\frac{1}{n} \|W_F\|_1 \cdot W_{b;C_2}^L \cdot h_B^{L-1}), \tag{10}$$

where $F_{W_b}^{C_1}(\hat{x}) - F_{W_b}^{C_2}(x)$ represents the difference between the two different outputs of DBNNs under noise perturbations and clean scenarios. Meanwhile, $W_{b;C_1}^L$ and $W_{b;C_2}^L$

denote the output layer of learnable parameters correspond to classes $C_1$ and $C_2$, respectively. Then, $h_B^{L-1}$ and $\hat{h}_B^{L-1}$ denote the intermediate results from the preceding $L-1$ layer under clean and noise scenarios, respectively.

**Theorem 4.2.** *For $L$-layer DBNNs against noise perturbations, we can derive the upper bound of robustness for the discrepancy between two classification outcomes (i.e., $C_1$ and $C_2$) as follows:*

$$F_{W_b}^{C_1}(\hat{x}) - F_{W_b}^{C_2}(x) \le (\prod_{l=1}^{L-1} \alpha_l \beta_l) \cdot \|W_{b;C_1}^L - W_{b;C_2}^L\|_1 \cdot$$

$$\prod_{l=1}^{L-1} \|(W_b^{L-l})^T\|_{1,\infty} \prod_{M=2}^{N} \|(W_b^M)^T\|_{1,\infty} \|\hat{h}_B^{M-1} - h_B^{M-1}\|_\infty, \tag{11}$$

*where $\hat{h}_B^M = I_b^M \otimes (W_b^M \cdots I_b^1 \otimes (W_b^1 \hat{x}))$ denotes the expansion of front $M$-layer DBNNs (i.e., $\beta I_b^M \otimes (\alpha W_b^M \cdots \beta I_b^1 \otimes (\alpha W_b^1 \hat{x}))$) can be contracted scaling factors (i.e., $\prod_{l=1}^{L-1} \alpha_l \beta_l$) by a 1-lipschitz continuous function. Here, $N$ denotes the total number of binarization layers that are perturbed by noise excluding the output layer of DBNNs. Then, $\|W_{b;C_1}^L - W_{b;C_2}^L\|_1$ represents the discrepancy in output layer of weights corresponding to the two categories (Suppose the DBNNs is judged to be of two different classes after applying noise) that are deemed distinct following noise perturbations. Thus, we constrain the scale factors and binary weights can effectively enhance the tightness of upper bound under noise perturbations.*

To facilitate the derivation of the bounds, we approximately omit the analysis of the residual structure. Nevertheless, we give the upper bound result of the Lipschitz constant for the residual block in the **Appendix**.

### 4.2. Tightness Analysis of $L_{1,\infty}$-norm Constraint

Firstly, the study (Shang et al., 2022) employs the Lipschitz continuity constrains to enhance the robustness of BNN-based models. However, it fails to provide a quantitative upper bound under environmental noise perturbations. According to the study (Shang et al., 2022), we can obtain an upper bound as follows:

$$\prod_{j=1}^{Q} L_{lip}^j < \prod_{j=1}^{Q} [\|W_b^j\|_2 \cdot \gamma^{j-Q}]^2, \tag{12}$$

where $\|w_b^j\|_2$ denotes the $L_2$ norm of binarized weights in the $j^{th}$ convolution layer, $Q$ denotes the total number of

binary convolution layers, $j \in [1, Q]$ and hyper-parameter $\gamma > 1$. Clearly, it is imperative to compare the quantitative relation between other norm constraints and the spectral norm constraints (Shang et al., 2022).

$$
\begin{aligned}
\|A\|_\infty^2 &= \max_i (\sum_{j=1}^n |a_{i,j}|)^2 \leq \sum_{i=1}^n (\sum_{j=1}^n |a_{i,j}|)^2 \\
&\leq n \sum_{i=1}^n \sum_{j=1}^n |a_{i,j}|^2 = n\|A\|_2^2 \,,
\end{aligned}
\tag{13}
$$

where $A$ denotes a standard matrix and $n$ represents the total number of rows in matrix. Obviously, we can get such an inequality relation $\|A\|_\infty \leq \sqrt{n}\|A\|_2$. The study (Liu et al., 2022) has focused on investigating the inequality relationship among different types of classical $L_1$, $L_2$, and $L_\infty$ norms for learning smooth neural functions. In contrast to the work (Liu et al., 2022), our aims to explore the quantitative relationship between the dual norm $L_{1,\infty}$ and the spectral norm $L_2$ for robustness noise perturbations bound in DBNNs. Firstly, the inequality relationship between the $L_{1,\infty}$ norm and the $L_\infty$ norm is as follows:

$$
\begin{aligned}
\|A^{\mathrm{T}}\|_{1,\infty} &= k \cdot \max \frac{\|A_i^{\mathrm{T}} x\|_\infty}{\|x\|_1} \leq k \cdot \max \frac{\|A_i^{\mathrm{T}}\|_\infty \cdot \|x\|_\infty}{\|x\|_1} \\
&\leq k \cdot \|A_i^{\mathrm{T}}\|_\infty \cdot \max \frac{\|x\|_\infty}{\|x\|_1} \,,
\end{aligned}
\tag{14}
$$

where $k$ denotes the number of dimensions of the matrix $A$. Based on the intermediate inequality (i.e., Eqn.13 and Eqn.14), we can derive the final bound inequality relation of one binary convolution layer in Eqn.15:

$$
\|A^{\mathrm{T}}\|_{1,\infty} \leq k \cdot \sqrt{n}\|A\|_2 \cdot \max \frac{\|x\|_\infty}{\|x\|_1} \,.
\tag{15}
$$

**Corollary 4.3.** *According to Eqn.12 and Eqn.15, we can determine the tightness ratio of $Q$ binary convolution layers between our study and previous work (i.e., LCR) (Shang et al., 2022) under noise perturbation as follows:*

$$
\frac{\prod_{j=1}^Q \|w_b^j\|_{1,\infty}}{\prod_{j=1}^Q L_{lip}^j} \leq \frac{k \cdot \sqrt{n}}{\|W_b^j\|_2 \cdot (\gamma^{j-Q})^2} \cdot \max \frac{\|x\|_\infty}{\|x\|_1} \,.
\tag{16}
$$

Here, the symbol $n$ denotes the dimension of the binary convolution weights. In fact, this is a definite value and does not change as the number of network layers increases. In addition, the upper bound of (Shang et al., 2022) increases exponentially and its value is much larger (due to the value of hyper-parameters $\gamma > 1$ in the LCR method) than the upper bound of our work, which means that the denominator of Cor.4.3 is always greater than the numerator, that is to say, the ratio is strictly less than 1. Theoretically, a compact

upper bound can improve the robustness of the model. With an increase in the network's layer count, the cumulative product of $L_{1,\infty}$-norm described in our method will also escalate; hence, we introduce a constraint coefficient ($\delta$) to regulate the constraint's strength in Eqn.18.

### 4.3. The $L_{1,\infty}$ Norm Constrain for Robust DBNNs

Given the multiple classification loss function $\ell$ on the image classification tasks, a series of randomly sampled image input matrices $\boldsymbol{X} = [x_1, \cdots, x_n]^{\mathrm{T}} \in \mathbb{R}^{n \times d}$ and ground truth $Y$, we can obtain a expected optimization objective as follows:

$$
\underset{X' \in \mathbb{R}^{n \times d}}{\arg\min} \, \ell(\boldsymbol{X} + X', Y) \,,
\tag{17}
$$

where $X'$ denotes the perturbed inputs according to Eqn.4. However, the optimization goal will deviate from the Eqn.17 due to the environmental perturbations that were not appeared during the training phase. The influence of noise perturbations during the inference phase on the robustness of DBNNs should be duly considered. Inspired by the theoretical analysis of Thm.4.2, we propose the $L_{1,\infty}$-norm constraint term to the objective function as the auxiliary robustness loss function $\mathcal{L}_p$:

$$
\mathcal{L}_p = \delta * \prod_{j=1}^Q \alpha \|W_b^j\|_{1,\infty} \,,
\tag{18}
$$

where the constrain coefficient $\delta$ (**The experiments in the Appendix discuss the effect of different coefficients on the performance.**) is utilized to ensure the maximization of a non-concave function (i.e., auxiliary robustness loss by using $L_{1,\infty}$-norm constrain) over a norm ball. Then, the coordination of the orders of magnitude between the our auxiliary robustness loss term and the classification loss term is crucial to prevent gradient disappearance or explosion rather than simple maximization. Here, $\|W_b^j\|_{1,\infty}$ denotes the $L_{1,\infty}$ norm of binarized weights in the $j^{th}$ convolution module, and $Q$ denotes the total number of binary convolution modules.

Given a classification-based objective function $\mathcal{L}_{total}$, we design a robustness loss function inside $\mathcal{L}_{total}$ through $L_{1,\infty}$-norm constrain as

$$
\mathcal{L}_{total} = \mathcal{L}_{mlc}(X, Y) + \mathcal{L}_p \,,
\tag{19}
$$

where $\mathcal{L}_{mlc}(X, Y)$ is the traditional cross-entropy loss function for multiple classification tasks. Here, the $X$ denotes the input image signal and $Y$ denotes the target label. Furthermore, it is crucial to ensure dimensional consistency in both the cross-entropy loss and our proposed $L_{1,\infty}$ constraint term. Please see the **Appendix** for an analysis of the impact of constraint coefficients $\delta$.

**The Automatic Training Framework for Various BNNs:** Based on the survey of BNNs (Yuan & Agaian, 2023), it has been observed that a common characteristics among various BNNs is their utilization of well-established deep backbone networks, such as ResNet18 and ResNet34. Thus, we provide the automatic training framework in Fig.2.

## 5. Experiments

### 5.1. Experimental Setups

**Datasets:** To comprehensively evaluate our proposed approach, we construct experiments on a series of popular BNNs on the large-scale bio-electricity (Roeder, 2022) classification, CIFAR-10, CIFAR-100 and ImageNet, Brain tumor MRI (Nickparvar, 2021) datasets with two common backbones (i.e., ResNet18 and ResNet34), see details of the setup in the **Appendix**.

**Noise Perturbations and Metrics:** We utilize a signal-to-noise ratio (SNR) level close to 50% for environmental noise. First, we apply the perturbations to the bio-electricity dataset is $\epsilon=[0.4, 0.8]$. Then, the range of environmental noise perturbations of Brain Tumor MRI dataset is $\epsilon=[0.05, 0.1]$. In addition, CIFAR-10, CIFAR-100 and ImageNet datasets, we select small perturbations $\epsilon=[1/255, 4/255]$ range of noise (refer to (Zhang et al., 2021)). Then, we utilize the test accuracy with noise metric to evaluate DBNNs. For details of clean test performance, please see the **Appendix**.

### 5.2. Experimental Results

**The Robustness of Our Method in Image Classification:** We conduct comprehensive experiments to evaluate the robustness of various DBNN-based models and our proposed method against environmental noise perturbations on popular image classification benchmarks (i.e., CIFAR-10 and CIFAR-100 datasets). To be fair, according to previous survey study (Qin et al., 2023), we have retrained several BNN-based models (i.e., BirealNet (Liu et al., 2018), ReActNet (Liu et al., 2020), DorefaNet (Zhou et al., 2016) and IRNet (Qin et al., 2020)) with our proposed method that utilize classical ResNet18 and ResNet34 backbone networks.

Specifically, we can draw several conclusions from the following aspects in Tab.2. (1) The experimental findings demonstrate that our approach significantly enhances the robustness of mainstream four BNN-based methods (Liu et al., 2018; 2020; Zhou et al., 2016; Qin et al., 2020) with two classical backbones, thereby emphasizing its universal applicability in our $L_{1,\infty}$ norm constraint. On the CIFAR-10 dataset, the application of our proposed method to IRNet significantly enhances its robustness test accuracy, achieving a remarkable increase of 2.23% to reach 93.64%. (2)

The proposed approach demonstrates a substantial enhancement in the robustness of models utilizing ResNet34 with BirealNet and IRNet when subjected to environmental noise perturbations on the two datasets. This means that our $L_{1,\infty}$ norm constraint strategy is more suitable for methods that have significant adjustments for learnable parameters. (3) The proportion of performance degradation under environmental noise perturbations are significantly higher on the CIFAR-100 dataset, which contains more complex image information, compared to the CIFAR-10 data set for both full-precision and binarized models. This implies that the general environmental noise perturbations have a greater impact on the decision-making process of the model in the intricate classification scenarios.

Finally, our work also provides a discussion on the selection of our designed constraint coefficients in Eqn.18. Due to space limitations, we provide several experiments of changing the hyperparameter for our constraint coefficients on enhancing the robustness of various BNN-based methods (Zhou et al., 2016; Liu et al., 2018; 2020; Qin et al., 2020; Tu et al., 2022) in the **Appendix**.

**The Robustness of Our Method in Bio-electrical and Brain tumor Classification:** To validate the generalization of our method against environmental noise perturbations, we examine the effect of $L_{1,\infty}$-norm constraint on the large-scale bio-electricity series (Roeder, 2022) and Brain tumor MRI classification (Nickparvar, 2021), as shown in Tab.3-4. We observe a 4% performance decline when replacing the binary convolution module with full-precision model. This means that for resources-constrained tasks, the robustness of the DBNN-based models are difficult to meet the requirements. For Bio-electricity task, our $L_{1,\infty}$-norm constraint strategy demonstrates a robustness improvement of up to 1% across the three binary convolution modules, thereby validating the efficacy of our strategy for diverse BNNs. For the brain tumor task, our strategy yields a 3-5% improvement in robustness for three DBNNs, thereby validating the effectiveness of the our method (See **Appendix** for details).

**Computational Overhead Analysis:** To mitigate the additional costs associated with training, we propose incorporating the $L_{1,\infty}$-norm constraint as an alternative solution, thereby eliminating the need for intricate approximation operations employed in the previous study (Shang et al., 2022). In Tab.5, we compare the actual time spent on training and testing time with the SOTA method (Shang et al., 2022) on the ImageNet dataset. Specifically, we propose eliminates the $L_2$-norm constraint module (Shang et al., 2022) in the inner embedding convolution operation, thereby reducing overhead from a model perspective. It is evident that our proposed method reduces training costs by 16% compared to LCR method (Shang et al., 2022). Thus, our robust training algorithm enables faster training speed for the DBNNs.

*Table 2.* Robustness comparison between our approach and popular BNN-based methods against environmental noise on the CIFAR-10 and CIFAR-100 datasets. Here, ↑ denotes the proposed method can improve the robustness of the existing BNN-based methods.

| Datasets | Scenarios | Backbones | Methods | | | | | | | | | |
|---|---|---|---|---|---|---|---|---|---|---|---|---|
| | | | FP32 (clean) | DoReFa | DoReFa+our | BiReal | Bireal+our | ReAct | ReAct+our | IRNet | IRNet+our |
| CIFAR-10 | With Noise | ResNet18 | 94.82 | 91.55 | 92.27↑ | 91.20 | 93.21↑ | 91.40 | 93.04↑ | 91.41 | **93.64**↑ |
| | | ResNet34 | 94.17 | 85.16 | 87.04↑ | 87.80 | **90.13**↑ | 87.87 | 89.00↑ | 87.88 | 88.31↑ |
| CIFAR-100 | With Noise | ResNet18 | 72.61 | 65.15 | 67.21↑ | 65.35 | 68.84↑ | 66.01 | 68.26↑ | 65.24 | **70.04**↑ |
| | | ResNet34 | 71.52 | 60.37 | 61.16↑ | 63.76 | **64.69**↑ | 60.48 | 63.81↑ | 60.61 | 61.71↑ |

*Table 3.* Robustness comparison between our strategy and three Binary Convolution structure against environmental noise on the bio-electricity series classification task.

| Performance | Methods | | | | | |
|---|---|---|---|---|---|---|
| | FP32(**clean**) | IRConv | IRConv+our | BirealConv | BirealConv+our | AdaBinConv | AdaBinConv+our |
| Test Acc. with Noise | 97.1 | 87.21 | **88.36**↑ | 88.66 | **89.51**↑ | 93.13 | **93.72**↑ |

*Table 4.* Test accuracy with noise comparison between our strategy and three DBNN-based baselines against environmental noise on the Brain tumor MRI dataset. Here, $w+$ denotes DBNN model with our strategy.

| Backbones | React | w+our | Dorefa | w+our | CycleBNN | w+our |
|---|---|---|---|---|---|---|
| ResNet18 | 76.77 | **83.87** | 84.84 | **86.91** | 83.36 | **85.46** |
| ResNet34 | 76.13 | **79.16** | 77.42 | **80.13** | 71.94 | **78.39** |

*Table 5.* Computational cost comparison between our and SOTA method on the ImageNet dataset. The notation (mm:ss) represents the unit of minutes and seconds of Epoch.

| Methods | Training(mm:ss) | Test (mm:ss) |
|---|---|---|
| Our | **56:08** | **4:49** |
| LCR (Shang et al., 2022) | 66:37 | 6:30 |

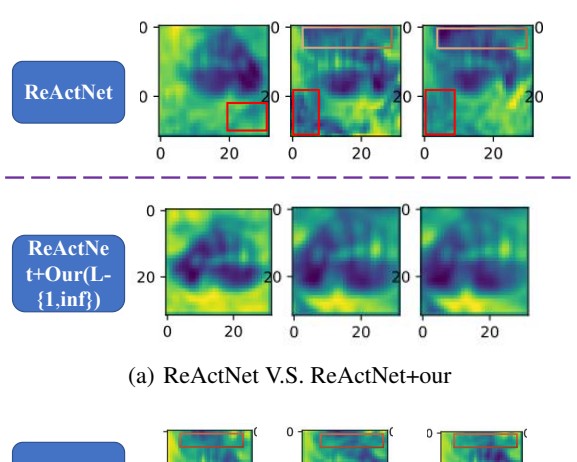

(a) ReActNet V.S. ReActNet+our

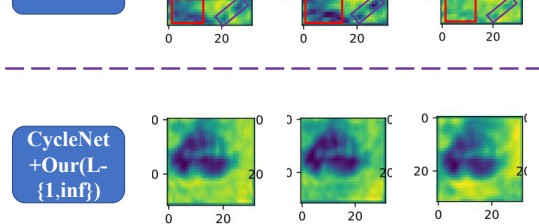

(b) CycleNet V.S. CycleNet+our

*Figure 4.* The visual feature maps between our method and two baselines (Liu et al., 2020; Fontana et al., 2024) under environmental noise perturbations on the CIFAR-100 dataset.

**The Visualization Case Study of Our Proposed Method for Enhancing Robustness on the CIFAR-100 Dataset:** To visually demonstrate the robustness of the proposed method against environmental noise on feature learning from images, we present a series of visualization cases on the CIFAR-100 dataset. In Fig.4, we randomly provide several figures of feature maps extracted by the first binary convolution layer before and after utilizing our proposed $L_{1,\infty}$-norm constraints term in the loss function. The primary observations in this case study are as follows:

(1) After applying our $L_{1,\infty}$-norm constraints, we observe that the baelines can attention towards the most crucial/center feature information under environmental noise perturbations. Specifically, our proposed method extracts image features with a heightened focus on individuals wearing glasses, in contrast to ReActNet's and CycleNet's broader emphasis on surrounding features (i.e., red/blue box).

(2) The feature maps of our proposed method have little color mixing zones around due to environmental noise perturbations and are clearer in the eyeglass frame compared to the several baseline models (Liu et al., 2020; Fontana et al., 2024) (i.e., the brown/blue box).

Thus, we can find that the proposed method has a positive effect on improving the robustness of DBNNs.

# 6. Conclusion

In this study, we answer that the robustness vulnerability of DBNNs (during inference) facing environmental noise is due to binary weights and scaling factors. To mitigate the adverse effects of environmental noise, we propose employing a $L_{1,\infty}$-norm constraint loss function in the training of DBNNs. Then, we quantitatively analyze the upper bound of noise perturbations which is more tighter than SOTA method. Experimental results demonstrate that our approach provides an effective strategy for DBNNs to mitigate environmental noise perturbations during inference across five different classification tasks.

## Acknowledgements

This work was supported in part by the National Natural Science Foundation of China (NSFC) under Grant U21A20513, Grant 62476157, Grant 62406179, the Stable Support Project of Shenzhen under Grant 20231120161634002 and the Shenzhen Science and Technology Program under Grant JCYJ20240813141417023.

## Impact Statement

This paper presents work whose goal is to advance the field of Machine Learning. There are many potential societal consequences of our work, none which we feel must be specifically highlighted here.

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

# A. Appendix

## A.1. Discussion of Differences with Existing Studies

In these remarkable verification researches (Baluta et al., 2019; Paulsen et al., 2020b;a; Mohammadinejad et al., 2021; Zhang et al., 2023; 2022), such researches only answer the Pass/Fail robustness verification questions (Jr. et al., 2024; Mistry et al., 2022; Henzinger et al., 2021) (i.e., The yellow box in Fig.1) of established models (i.e., whether the well-trained model is pass or fail). In the event that the established model fails to pass formal verification, it becomes imperative to iterate through the training process once again. This implies that the aforementioned process exhibits sub-optimal efficiency in real-time task execution on edge devices.

Many papers on formal verification of quantized/binarized NNs (Baluta et al., 2019; Paulsen et al., 2020b;a; Mohammadinejad et al., 2021; Zhang et al., 2023), which can verify a given NN against a specific property, e.g., local robustness verification and maximum robustness radius computation in QVIP (Zhang et al., 2022). Our work aims to produce a more robust Binary NN, with empirical evaluation based on a test dataset, similar to several comparison baselines (Qin et al., 2020; 2023; Shang et al., 2022). Note that the "upper bound" derived in our paper refers to the Lipschitz constant of the NN, a metric that measures the maximum rate of change of the network's output with respect to changes in its input, but we do not address the formal verification problem (as we mentioned above). In real-world scenarios, the input noise may be unpredictable and encompass various types of noise unseen during the training phase, and without any upper bound on its radius or norm. Our work proposes a robust training algorithm for enhancing robustness against general input noise perturbations during the testing phase, and empirical experiments confirm its effectiveness.

The reason that our work and comparison baselines do not use formal verification is due to the well-known scalability issue (in survey study (Meng et al., 2022)) that limits the size of NNs that can be formally verified with both soundness and completeness. Subsequently, numerous comprehensive analyses have been conducted to explore the intricate relationship between robustness (Shkolnik et al., 2020) and quantization (Lin et al., 2021). Thus, it is worth noting that quantization/binarization does not hurt the robustness of the model when compared to its full-precision counterpart.

## A.2. Proof of Theorem 1

Firstly, we employ several matrix norm inequalities to determine the upper bound of this discrepancy by using the rule from an example in Eqn.7. Then, we have

$$
\begin{aligned}
&\overset{a}{\leq} \|W_{b;C_1}^L - W_{b;C_2}^L\|_1 \\
&\cdot \|\{\frac{1}{n}\|I_F\|_1 \cdot I_b(\frac{1}{n}\|W_F\|_1 \cdot W_b^{L-1} \cdot \hat{h}_B^{L-2})\} \\
&- \{\frac{1}{n}\|I_F\|_1 \cdot I_b(\frac{1}{n}\|W_F\|_1 \cdot W_b^{L-1} \cdot h_B^{L-2})\}\|_\infty \,.
\end{aligned}
\tag{20}
$$

The establishment of inequality $\overset{a}{\leq}$ is attributed to the rule of Eqn.9 and the problem definition of Def.4.1. Furthermore, we have

$$
\overset{b}{\leq} \|W_{b;C_1}^L - W_{b;C_2}^L\|_1 \cdot \|W_b^{L-1}(\hat{h}_B^{L-2} - h_B^{L-2})\|_\infty \,.
\tag{21}
$$

The establishment of inequality $\overset{b}{\leq}$ is the contractive property of binary activation function and scaling factors (i.e., $\alpha$ and $\sigma$) combined with triangle inequality. To establish the tightness of the upper bound on robustness of DBNNs, it is crucial to thoroughly analyze the intermediate conclusion in Cor.A.1 and subsequently conclude the proof for Thm.4.2.

The contractive inequality in the subsequent step is derived through a crucial Cor.A.1, which constitutes the pivotal aspect of the upper bound derivation in our work.

**Corollary A.1.** *Given a standard matirx $A$ and a vector $\vec{z}$, we have $\|A\vec{z}\|_\infty \leq \|A^T\|_{1,\infty}\|\vec{z}\|_\infty$.*

Let $A = [a_{i,j}]$ where $i \in [1, \cdots, m], j \in [1, \cdots, n]$ and $\vec{z} = [z_i]_{1 \leq j \leq n}^T$ and $I_{m \times m} = [\vec{t_1}, \vec{t_2}, \cdots, \vec{t_m}]$. Here, $[\vec{t_k}]_i$ represents

the $i^{th}$ term of $\vec{t}_k$. Then, we have the following derivation ($\sum_{j=1}^{n} |a_{k,j}|$):

$$\|A^T\|_{1,\infty} = \max_{\vec{z}\neq\vec{0}} \frac{\|A^T\vec{z}\|_1}{\|\vec{z}\|_\infty} = \max_{\vec{z}\neq\vec{0}} \frac{\sum_{j=1}^{n} |\sum_{i=1}^{m} a_{i,j}z_i|}{\max_{1\leq i\leq m} \{|z_i|\}}$$

$$\geq \frac{\sum_{j=1}^{n} |\sum_{i=1}^{m} a_{i,j}[\vec{t_k}]_i|}{\max_{1\leq i\leq m} \{|[\vec{t_k}]_i|\}} \ . \tag{22}$$

In order not to lose generality, it is obvious that the number of rows (i.e., $k$) is arbitrary, and thus we have:

$$\|A^T\|_{1,\infty} \geq \max_{1\leq k\leq m} \sum_{j=1}^{n} |a_{k,j}| \geq \max_{1\leq k\leq m} |\sum_{j=1}^{n} a_{k,j}| \ . \tag{23}$$

Finally, the inequality of Cor.A.1 can be proved as follows:

$$\|A\vec{z}\|_\infty = \max_{1\leq i\leq m} \{|\sum_{j=1}^{n} a_{i,j}z_j|\} \leq \max_{1\leq i\leq m} \{|\sum_{j=1}^{n} a_{i,j}| \cdot \max_{1\leq j\leq n} |z_j|\}$$

$$\leq \max_{1\leq i\leq m} \{|\sum_{j=1}^{n} a_{i,j}|\} \cdot \max_{1\leq j\leq n} \{|z_j|\} \leq \|A^T\|_{1,\infty} \|\vec{z}\|_\infty \ , \tag{24}$$

where the above inequality is established by using the matrix norm consistency theorem.

Utilizing the Cor.A.1, we continue to give the proof of the upper bound from Thm.4.2 as follows:

$$\overset{c}{\leq} \|W_{b;C_1}^L - W_{b;C_2}^L\|_1 \cdot \|(W_b^{L-1})^T\|_{1,\infty}$$

$$\cdot \{\|\frac{1}{n}\|I_F\|_1 \cdot I_b \cdot (\frac{1}{n}\|W_F\|_1 \cdot W_b^{L-2} \cdot \hat{h}_B^{L-3})$$

$$- \frac{1}{n}\|I_F\|_1 \cdot I_b \cdot (\frac{1}{n}\|W_F\|_1 \cdot W_b^{L-2} \cdot h_B^{L-3})\|_\infty\} \ . \tag{25}$$

Subsequently, the contractive property of the 1-lipschitz continuous function is utilized to further process several scaling factors and binary activation functions in the Eqn.25, thereby organizing the continued product terms associated with the input of the initial layer. Based on the key corollary (i.e., Cor.A.1), we can derive the following two inequalities:

$$\overset{d}{\leq} \|W_{b;C_1}^L - W_{b;C_2}^L\|_1 \cdot \|(W_b^{L-1})^T\|_{1,\infty}$$

$$\cdots \|(W_b^{N+1})^T\|_{1,\infty} \cdot \|W_b^N \cdot (\hat{h}_B^{N-1} - h_B^{N-1})\|_\infty \ ,$$

$$\overset{e}{\leq} (\prod_{l=1}^{L-1} \alpha_l\beta_l) \cdot \|W_{b;C_1}^L - W_{b;C_2}^L\|_1 \cdot \prod_{l=1}^{L-1} \|(W_b^{L-l})^T\|_{1,\infty} \tag{26}$$

$$\cdot \prod_{M=2}^{N} (\|(W_b^M)^T\|_{1,\infty} \cdot \|\hat{h}_B^{M-1} - h_B^{M-1}\|_\infty) \ .$$

Finally, the conclusion of Thm.4.2 can be proven.

The previous study (Gouk et al., 2021) has examined the disparity of the Lipschitz constants between the residual module and the convolutional layer, albeit acknowledging that calculating this aspect of DBNNs is relatively intricate. For the sake of theoretical accuracy, we provide an upper bound for the residual connection module through subsequent corollary.

Firstly, in the standard ResNet backbone, the computation process of residual module ($\phi^{res}(\mathbf{x})$) can be described as follows:

$$\phi^{res}(\mathbf{x}) = \mathbf{x} + (\phi_{q+p} \circ \ldots \circ \phi_{q+1})(\mathbf{x}) \ , \tag{27}$$

where $p$ represents the maximum number of layers to span in a residual connection module. The existing study has given the following corollary through Lipschitz continuity:

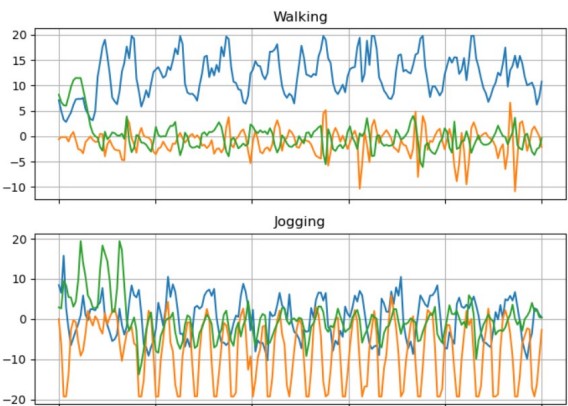

*Figure 5.* The example of data point corresponding to distinct human actions.

**Corollary A.2** (from (Gouk et al., 2021))**.** *We can find that the Lipschitz constant of a residual module is bounded by* $L(\phi^{res}) \leq 1 + \prod_{q=1}^{q+p} L(\phi_i)$ , *where* $L(\phi_i)$ *represents the Lipschitz constant of the* $i$-*th module of the neural network.*

Based on the conclusion of Cor.A.2 and Eqn.26, we can add residual module of the upper bound to the last term of the right-hand side in Eqn.28 as follows:

$$
\begin{aligned}
\overset{e}{\leq} \ & (\prod_{l=1}^{L-1} \alpha_l \beta_l) \cdot \|W_{b;C_1}^L - W_{b;C_2}^L\|_1 \cdot \prod_{l=1}^{L-1} \|(W_b^{L-l})^T\|_{1,\infty} \\
& \cdot \prod_{M=1}^{N} (\|(W_b^M)^T\|_{1,\infty} \cdot \|\hat{h}_B^{M-1} - h_B^{M-1}\|_\infty) \\
& \cdot k(1 + \prod_{i=M+1}^{M+p} (\|(W_b^i)^T\|_{1,\infty} \cdot \|\hat{h}_B^{i-1} - h_B^{i-1}\|_\infty)) ,
\end{aligned}
\tag{28}
$$

where $k$ denotes the total number of the residual module. In the case of employing ResNet18 as the backbone, the residual connections encompass a total of four BNN module layers, wherein each layer comprises two sets of residual modules. Consequently, the value of $k$ is determined to be 8.

### A.3. Detailed Experimental Setup

**Datasets:** (1) The Bio time series dataset (Kaggle project) stems from the esteemed neural networks and cognitive models course offered by the MAI program at FHWS (Roeder, 2022). The dataset comprises data from 51 distinct subjects, each identified uniquely within the train and test sets by their IDs ranging from 1600 to 1650. Total 120K data points encapsulate low-level, high-frequency (20 Hz, or every 50 milliseconds) time-series sensor readings, specifically the x, y, and z coordinates recorded by the smartwatch's accelerometer. After introducing general input noise, we show the bio-metric signals corresponding to distinct human actions in Fig.5. (2) The CIFAR-10 is the most popular image classification dataset, which consists of 50,000 training samples and 10,000 testing samples of size 32 x 32 divided into 10 image classes. (3) The CIFAR-100 dataset is similar to CIFAR-10 dataset, with the exception that it consists of 100 classes, each containing 600 images. (4) The ImageNet dataset provide 1.2 million training samples and 50,000 validation samples, distributed across a total of 1,000 distinct classes to facilitate the exploration of complex research tasks.

**Experimental Setup Details: Datasets:** Brain tumors can be classified as either malignant or benign. The growth of both types of tumors can lead to an increase in intracranial pressure. Early detection and accurate classification of brain tumors is a critical research area within medical imaging, which significantly aids in selecting the most appropriate treatment strategy to save patients' lives. This dataset (Nickparvar, 2021) comprises 7,023 human brain MRI images categorized into four classes: glioma, meningioma, no tumor, and pituitary tumor. The dataset is divided into training, validation, and test sets following a conventional 7:2:1 ratio. **Training Details:** Firstly, we validated several DBNN-based models with two common backbones (i.e., ResNet34 and ResNet18) on some image classification datasets, namely ABCNet (Lin et al., 2017),

BirealNet (Liu et al., 2018), ReActNet (Liu et al., 2020), DorefaNet (Zhou et al., 2016), IR-Net (Qin et al., 2020), AdaBin (Tu et al., 2022) and CycleBNN (Fontana et al., 2024). The rationale for selecting the aforementioned baselines lies in their widespread recognition and high citations within the research community. To validate the robustness of the DBNN-based training algorithm by using various matrix norm constraint strategies, we compare our proposed method with the LCR method (Shang et al., 2022) on the ImageNet dataset. For the task of bio-electricity series classification, we opted to replace the 2D convolution module in the full-precision model with binary convolution modules (i.e., IRConv (Qin et al., 2020), BirealConv (Liu et al., 2018), and AdaBinConv (Tu et al., 2022)) that are currently popular to align with the requirements of a shallow network, as opposed to deep networks used for image classification tasks. We also evaluate the clean test accuracy of our method with several baselines on the CIFAR-10 and CIFAR-100 datasets.

All experiments are executed on a Linux server (i.e., Ubuntu 18.04.6 LTS) with one RTX 3090 GPUs. Specifically, we train all baselines and our method with SGD optimizer by using common hyper-parameters (i.e., momentum=0.9, weight decay=$1e-4$) under PyTorch1.13-GPU library on several classification benchmarks, namely CIFAR-10, CIFAR-100, ImageNet, Bio-electricity series and Brain Tumor MRI datasets. For image classification tasks, the batch size is set to 128 and then epoch is set to 400. Furthermore, the initial learning rate is $1e-1$ on the three image benchmark datasets. In addition, we follow the popular cosine of learning rate decay strategy in the survey (Qin et al., 2023). To understand the process of applying environmental noise in the manuscript, we give the pseudo-code in Algo.1.

---

**Algorithm 1** Model with Environmental Noise Perturbations during Inference

---

**Input:** Sample $x$, size $m$ from Dataset $D$
**Require:** Dateset $D$ for Test.

1: **for** size of test set $D$ **do**
2:     Sample $x$ from $D$.
    //Apply the environmental noise for each test sample.
3:     **for** $j = 1$ **to** size of test samples $N$ **do**
4:       **for** $i = 1$ **to** total number of pixel noise points $P$ **do**
5:         $randx = np.random.randint(1, h-1)$ //Random height position ($h$).
6:         $randy = np.random.randint(1, w-1)$ //Random width position ($w$).
7:         **if** $np.random.random() \leq 0.5$ and $j \leq N$ **then**
8:           $x[j, :, randx, randy] = x[j, :, randx, randy] + \epsilon$ //Use function $random(\cdot)$ to generate random noise perturbations $\epsilon$.
9:         **end if**
10:       **end for**
11:     **end for**
12: **end for**
13: **Return:** The test sample with environmental noise perturbations during inference.

---

**Evaluation:** Following the evaluation method of a previous robustness study (Zhang et al., 2021), we employ two metrics, namely 'No Noise' (representing clean test accuracy) and 'With Noise' (indicating test accuracy under environmental noise with high intensity signal-to-noise ratio (SNR) set at nearly 50%) to evaluate the performance of all BNN-based methods.

### A.4. Additional Experiments

**Clean Test Accuracy Comparison:** The study (Shang et al., 2022) claims that the norm constraints by using the Lipschitz continuous to enhance test accuracy of BNN-based models under the clean scenarios. To verify such capability, we conduct several experiments to evaluate our strategy on the CIFAR-10 dataset, as shown in Tab.6.

In particular, we choose the IRNet (Qin et al., 2020) as our fundamental BNN-based model for conducting comparative experiments due to its demonstrated effectiveness in the field of BNNs, as emphasized in several studies (Tu et al., 2022; Shang et al., 2022). According to the experimental setups of the study (Shang et al., 2022), we employ ResNet18 and ResNet20 as the two backbone networks. Then, we construct experiments by employing our proposed method for IRNet, denoted as IRNet+our. For the ResNet18 backbone, the IRNet+our method has significantly enhanced the clean test accuracy of IRNet method, surpassing the performance achieved by the LCR method. Then, the proposed method demonstrates enhanced test accuracy for the IRNet method, even when the depth of the backbone network is increased to ResNet20.

*Table 6.* Performance comparison between the our method and several SOTA methods on the clean CIFAR-10 dataset. Here, the (W/A) denotes the weights and activations of Bit-width. Here, notation * indicates that the activations are in full precision.

| Backbones | Methods | Bit-width (W/A) | Clean Test Accuracy |
|---|---|---|---|
| ResNet18 | Full Precision (Qin et al., 2023) | 32/32 | 94.82 |
| | IRNet (Qin et al., 2020) | 1/1 | 91.50 |
| | LCR (Shang et al., 2022) | 1/1 | 91.80 |
| | IRNet+our | 1/1 | **94.06**↑ |
| ResNet20 | Full Precision (Qin et al., 2023) | 32/32 | 91.99 |
| | IRNet (Qin et al., 2020) | 1/1 | 85.50 |
| | LCR (Shang et al., 2022) | 1/1 | 86.00 |
| | IRNet+our | 1/1 | **88.19**↑ |
| ResNet20 | IRNet* (Qin et al., 2020) | 1/32 | 90.80 |
| | LCR* (Shang et al., 2022) | 1/32 | 91.20 |
| | IRNet+our* | 1/32 | **91.41**↑ |

*Table 7.* Investigation of the Influence of our constraint coefficient on the robustness of various BNNs against environmental noise on the CIFAR-10 dataset.

| Methods | Backbones | Noise | Coefficient | Test Acc. |
|---|---|---|---|---|
| AdaBin | ResNet34 | Noise | N/A | 87.55 |
| AdaBin+our | ResNet34 | Noise | 5e-3 | 87.85 ↑ |
| AdaBin+our | ResNet34 | Noise | 5e-2 | **88.08** ↑ |
| DoReFa | ResNet34 | Noise | N/A | 85.16 |
| DoReFa+our | ResNet34 | Noise | 5e-3 | **87.04** ↑ |
| DoReFa+our | ResNet34 | Noise | 5e-2 | 86.92 ↑ |
| ReAct | ResNet34 | Noise | N/A | 87.87 |
| ReAct+our | ResNet34 | Noise | 5e-3 | **89.00** ↑ |
| ReAct+our | ResNet34 | Noise | 5e-2 | **89.00** ↑ |
| Bireal | ResNet34 | Noise | N/A | 87.80 |
| Bireal+our | ResNet34 | Noise | 5e-3 | 90.13 ↑ |
| Bireal+our | ResNet34 | Noise | 5e-2 | **90.40** ↑ |
| IRNet | ResNet34 | Noise | N/A | 87.88 |
| IRNet+our | ResNet34 | Noise | 5e-3 | 87.93 ↑ |
| IRNet+our | ResNet34 | Noise | 5e-2 | **88.31** ↑ |

Meanwhile, we can observe a slight improvement in the IRNet+our method than the LCR method when the activation of the model is 32-bit. Thus, the above empirical results demonstrate that the application of our $L_{1,\infty}$-norm constraints can enhance the performance of BNN-based models in clean scenarios, surpassing the effectiveness of spectral norm (Shang et al., 2022) constraints.

**The Influence of Our Constraint Coefficient on Robustness:** Clearly, the distinct design of the BNN-based methods imply that our provided general noise disturbance bound constraints can enhance the environmental noise robustness of various BNNs to varying degrees. To further imporve robustness of such BNN-based methods (Zhou et al., 2016; Liu et al., 2018; 2020; Qin et al., 2020; Tu et al., 2022), we conducted several quantitative experiments for the constraint coefficient on the CIFAR-10 dataset, as illustrated in Tab.7. Here, the upward arrow in Tab.7 denotes the enhanced robustness of several BNN-based methods (Zhou et al., 2016; Liu et al., 2018; 2020; Qin et al., 2020; Tu et al., 2022) by using our $L_{1,\infty}$ constraints. Specifically, the ReAct method (Liu et al., 2020) exhibits insensitivity to variations in constraint coefficients following the application of our approach. For the DoReFa method (Zhou et al., 2016), a smaller constraint coefficient (i.e., $5e-3$) demonstrates enhanced robustness. Conversely, for the remaining methods (Liu et al., 2018; Qin et al., 2020; Tu et al., 2022) utilizing our $L_{1,\infty}$ norm constraint term, a larger coefficient demonstrates enhanced robustness. In addition, our proposed $L_{1,\infty}$ norm constraint term for the latest AdaBin method (Tu et al., 2022) should be carefully, as it is highly susceptible to gradient calculation anomalies (i.e., NAN) when the coefficients are in the $1e-1$.

**The Trade-offs between Clean Test Accuracy and Robustness in Our $L_{1,\infty}$-norm constraint method:** It is worth

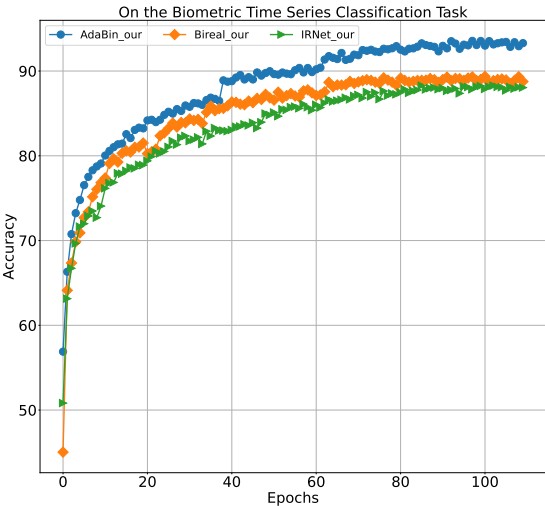

*Figure 6.* The accuracy curves on bio-electricity time series dataset.

*Table 8.* Analysis of the clean test accuracy of our proposed $L_{1,\infty}$-norm constraint strategy with baseline models on the Brain Tumor MRI dataset

| Methods | Test Accuracy(clean) |
|---|---|
| ResNet18(FP32) | 92.58 |
| ResNet34(FP32) | 90.32 |
| ResNet18-Cyclebnn | 84.84 |
| ResNet34-Cyclebnn | 72.35 |
| ResNet18-Cyclebnn+our | **89.98** |
| ResNet34-Cyclebnn+our | **82.26** |

noting that a certain degree of under-fitting may occur in some full precision CNN-based models due to the utilization of regularization techniques (Wei & Zhao, 2024). In other words, the model's accuracy on clean test data is observed to be lower than its performance in noisy scenarios. To investigate the impact of the aforementioned phenomenon on the DBNN-based model, a series of experiments is presented in Tab.10. Specifically, it can be observed that only a subset of BNNs exhibit a marginal decline in terms of clean test accuracy when the proposed $L_{1,\infty}$ constraint loss function is employed. Our method achieves a remarkably low loss of 0.06 for the ABC+our, specifically when employing the ResNet18 backbone network. Conversely, IRNet exhibits a slightly performance loss of 0.66 with the proposed method. The above experiments demonstrate that our proposed method strikes an appropriate balance between clean test accuracy and robustness. In Tab.8, we can find that the clean test accuracy of our proposed $L_{1,\infty}$-norm constraint strategy with baseline models on the Brain Tumor MRI dataset. The experimental results show that our proposed strategy can also improve the clean classification accuracy of the model during inference.

**Investigation of the impact of the $L_{1,\infty}$-norm constraint approach on the robustness of the full precision model:** The $L_{1,\infty}$-norm constraints are commonly believed to enhance the robustness of full-precision models. However, due to the quantization operation, there exists a certain disparity between the calculation processes of deep CNNs and BNNs. Hence, we present a case analysis (on the CIFAR-100 dataset) in Tab.9. Specifically, we observe that the proposed method enhances robustness (i.e., increasing 0.5%) in deeper structures (i.e., ResNet34), while leading to performance degradation in shallower structures (i.e., ResNet18). Firstly, several experimental results in Tab.2 and Tab.9 show that our proposed method is specifically for improving the robustness of DBNN-based models. Then, the above improving discrepancy primarily stems from the scaling factor and XNOR operation employed in DBNN-based models, which are not present in full precision models. Consequently, the robustness improvement offered by our approach is limited for full-precision models with specific ResNet backbone.

**The Effectiveness of Different $L$-norm Constraints on the Robustness of DBNNs:** To comprehensive understanding the effectiveness of various $L$-norm constraints for the robustness of DBNNs against environmental noise, we conducted

*Table 9.* Analysis of the robustness effect of our proposed $L_{1,\infty}$-norm constraint for full precision model on the CIFAR-100 dataset

| Methods | Test Acc.with Noise |
|---|---|
| ResNet18 | 71.96 |
| ResNet18+our | 69.33↓ |
| ResNet34 | 71.23 |
| ResNet34+our | **71.73↑** |

*Table 10.* The Trade-offs between Clean Test Accuracy and Robustness of our $L_{1,\infty}$ constraint method on the CIFAR100 dataset

| Scenarios | Backbones | Methods | | | | | |
|---|---|---|---|---|---|---|---|
| | | DoReFa+our | ABC+our | Bireal+our | ReAct+our | IRNet+our | CycleBNN+our |
| Clean | ResNet18 | 67.59 | 69.17↓ | 68.96 | 68.20↓ | 69.30↓ | 67.35 |
| Noise | ResNet18 | 67.21 | 69.23 | 68.84 | 68.26 | 70.04 | 67.29 |
| Clean | ResNet34 | 61.18 | 63.34↓ | 64.11↓ | 64.05 | 61.46↓ | 68.90 |
| Noise | ResNet34 | 61.16 | 63.47 | 64.69 | 63.81 | 61.71 | 68.36 |

relevant experiments on the CIFAR-10 and ImageNet in Tab.11. To begin with, we adopt similar experimental setups from the LCR method (Shang et al., 2022), wherein the ResNet18 backbone network is trained using the BirealNet model for LCR and our approaches on the ImageNet dataset. To verify the extensibility of our bounds, a deeper ResNet34 is provided as the backbone network. Recently, some $L$-norm constraints have been proposed to improve the robustness of CNN-based model. Thus, we construct experiments to analyze how much they improve the robustness of DBNNs. Specifically, $L_{lip}$ represents the spectral norm constraint after the approximate operations of the LCR method (Shang et al., 2022). Then, the special norm constraints $L_2$ and $L_\infty$ are derived from the studies (Li et al., 2019) and (Kanai et al., 2020), respectively.

In Tab.11, experimental results demonstrate that our proposed $L_{1,\infty}$ norm constraint can enhance robustness of the Bireal method by 0.96% compared to the LCR method (Shang et al., 2022) when applied to the ImageNet dataset. Meanwhile, the experimental results show that our $L_{1,\infty}$ norm constraint can improve robustness by 1.98% compared to the Bireal (Liu et al., 2018) on the CIFAR-10 dataset. Furthermore, the robustness of BNNs by using our $L_{1,\infty}$-norm constraint surpasses the $L_1$, $L_2$ and $L_\infty$ norm constraints on the CIFAR-10 and ImageNet datasets. This means that our $L_{1,\infty}$-norm constraints can bring effective robustness improvement under environmental noise perturbations in the inference phase to DBNNs. In addition, another advantage of our $L_{1,\infty}$-norm constraint is that it can also improve the robustness of existing constraint methods, such as the 'LCR_Bireal+our ($L_{1,\infty}$)' in Tab.11.

**The Effectiveness of Our Constraints on the Distribution of Learnable Parameters:** An intriguing inquiry pertains to the impact of matrix norm constraints, such as $L_2$ or $L_{1,\infty}$, on the learnable parameters within DBNNs. Revealing this inquiry, we employ visual method to compare our proposed method with IRNet (Qin et al., 2020) on the CIFAR-10 dataset. In Fig.7(a) and Fig.7(b), the learnable parameter distribution of IRNet (Qin et al., 2020) exhibits a unimodal form, which may potentially limit the model's capacity to capture complex features and consequently hinder its performance. Then, the learnable parameters distribution of the IRNet using our $L_{1,\infty}$ norm constraint, exhibits a smoother histogram profile compared to IRNet, as illustrated in Fig.7(c) and Fig.7(d). Meanwhile, the study (Liu et al., 2021) has been shown that

*Table 11.* Robustness comparison of our and SOTA methods using different $L$-norm constraints on the CIFAR-10 and ImageNet datasets.

| Datasets | Methods with $L$-norm Constraints | Backbones | Scenario | Test Accuracy with Noise |
|---|---|---|---|---|
| ImageNet | LCR_Bireal+$L_{lip}$ | ResNet18 | Noise | 57.27 |
| ImageNet | LCR_Bireal+$L_1$ | ResNet18 | Noise | 57.84 |
| ImageNet | LCR_Bireal+$L_2$ (Li et al., 2019) | ResNet18 | Noise | 57.98 |
| ImageNet | LCR_Bireal+$L_\infty$ (Kanai et al., 2020) | ResNet18 | Noise | 58.07 |
| ImageNet | Bireal+our ($L_{1,\infty}$) | ResNet18 | Noise | **58.23** |
| ImageNet | LCR_Bireal+our ($L_{1,\infty}$) | ResNet18 | Noise | **58.35** |
| CIFAR-10 | Bireal+$L_1$ | ResNet34 | Noise | 88.15 |
| CIFAR-10 | Bireal+$L_2$ (Li et al., 2019) | ResNet34 | Noise | 88.34 |
| CIFAR-10 | Bireal+$L_\infty$ (Kanai et al., 2020) | ResNet34 | Noise | 88.48 |
| CIFAR-10 | Bireal+our ($L_{1,\infty}$) | ResNet34 | Noise | **90.13** |

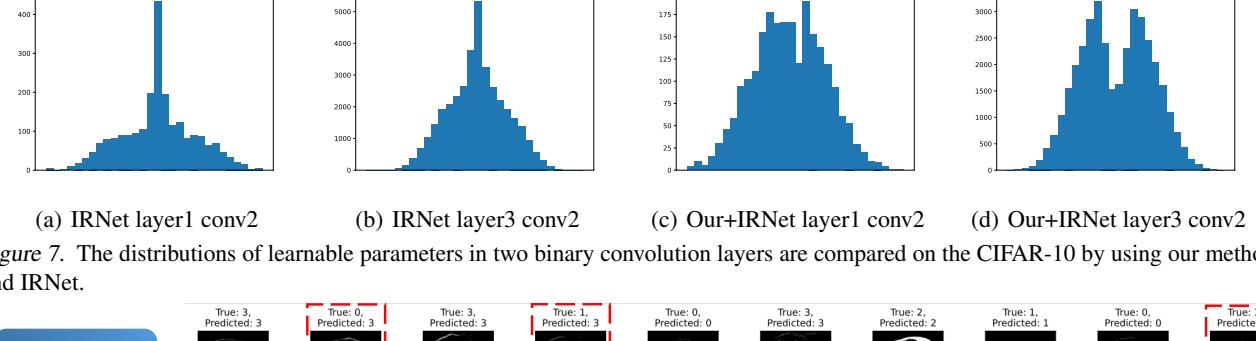

|(a) IRNet layer1 conv2 | (b) IRNet layer3 conv2 | (c) Our+IRNet layer1 conv2 | (d) Our+IRNet layer3 conv2 |

*Figure 7.* The distributions of learnable parameters in two binary convolution layers are compared on the CIFAR-10 by using our method and IRNet.

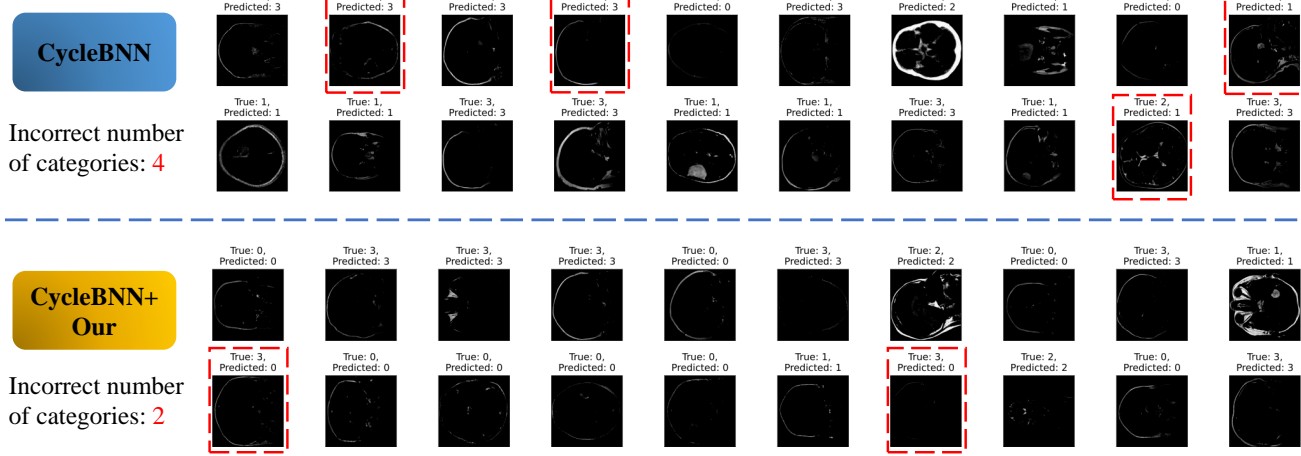

*Figure 8.* Visual comparison between the proposed strategy with CycleBNN (i.e., Yellow rectangle with rounded corners) and standard CycleBNN model (Fontana et al., 2024) on the brain tumor test dataset under environmental noise perturbations.

restoring the diversity of learnable parameter distribution by special optimization settings can improve the performance of BNNs. Our $L_{1,\infty}$ norm constraint introduces more peaks in Fig.7(c) and Fig.7(d), indicating an positive impact of our approach on restoring the diversity of learnable parameters.

**The Visualization of Our Strategy with Several Binary Convolution Modules on the Bio-electrical (Roeder, 2022) dataset:** To show the advantages of the proposed strategy over several popular binary convolution modules against general environmental noise perturbations, we present its accuracy rate iterated over epochs in a visual manner as shown in Fig.6.

**The Visualization of Our Strategy with a CycleBNN (Fontana et al., 2024) model in the Brain tumor image classification (Nickparvar, 2021) dataset:** To validate the effectiveness of the proposed constraint training strategy in enhancing robustness for a real-world resource-constrained task, we select the open-source brain tumor MRI task. Visual comparison results as shown in Fig.8. Specifically, we randomly chose a batch of 20 samples perturbed by environmental noise to visually demonstrate the improvement. Then, the model employs ResNet18 as the backbone network, and the most recent CycleBNN (Fontana et al., 2024) serves as the BNN algorithm. The environmental noise level is set within the range of 0.05 to 0.1, which closely mimics the environmental noise characteristics of MRI image data. Additionally, the SNR is enhanced by 50%. Here, the well-trained CycleBNN by using our proposed $L_{1,\infty}$-norm constraints (i.e., Yellow rectangle with rounded corners) exhibits an un-robust rate in the inference that is half that of the CycleBNN model (i.e., Blue rectangle with rounded corners). In addition, our strategy achieves 90% test accuracy in inference tasks involving random environmental noise, which is sufficient to meet the practical requirements for assisting primary care physicians in diagnostic decision-making.

**Analysis of Additional Computational Overhead:** Specifically, we have evaluated the training cost of a series of BNN-based methods on CIFAR-100 and Brain Tumor datasets. We have reported the average training time of each epoch and total test time. According to the results presented in the main paper and two table below, it can be observed that an appropriately designed penalty term not only enhances model robustness but also significantly improves training efficiency. In addition, the overhead in the test phase also encompasses the time required for adding environmental noise. The detailed computational overhead are presented Tab. 12-13.

**Analysis of the PGD Attack for DBNNs:** We opted for a standard training and testing phase to introduce PGD attacks,

*Table 12.* Computational cost comparison between popular BNN-based methods and our on the CIFAR-100 dataset.

| Methods (ResNet18 backbone) | Training Time/Epochs (s) | Total Test Time (s) |
|---|---|---|
| CycleBNN | 24 | 43 |
| CycleBNN+Our | **21** | **42** |
| IRNet | 13 | 32 |
| IRNet+Our | **12** | **31** |
| Dorefa | 11 | 33 |
| Dorefa+Our | 11 | 33 |
| React | 14 | 33 |
| React+Our | **13** | 33 |
| Bireal | 13 | 30 |
| Bireal+Our | **12** | 30 |
| ABCNet | 13 | 41 |
| ABCNet+Our | **12** | 41 |

*Table 13.* Computational cost comparison between BNNs and our on the Brain tumor dataset.

| Methods (ResNet18 backbone) | Training Time/Epochs (s) | Total Test Time (s) |
|---|---|---|
| CycleBNN | 13 | 8 |
| CycleBNN+Our | **11** | 8 |
| Dorefa | 11 | 3 |
| Dorefa+Our | **10** | 3 |
| React | 6 | 4 |
| React+Our | **5** | 4 |

*Table 14.* PGD performance comparison between CycleBNN and our on the CIRFA-100 dataset.

| Models | Test PGD |
|---|---|
| CycleBNN | 17.78 |
| CycleBNN+Our | **18.38** |

*Table 15.* Ablation study 1: The contribution of constrained binary weights and scaling factors to improving environmental noise robustness of DBNNs on the CIRFA-100 dataset. The w/o indicates that the constraint on the scaling factors has been removed.

| Backbone | Dorefa+Our | Dorefa+Our+w/o | React+Our | React+Our+w/o | ABC+Our | ABC+Our+w/o | IRNet+Our | IRNet+Our+w/o | Bireal+Our | Bireal+Our+w/o | CycleBNN+Our | CycleBNN+Our+w/o |
|---|---|---|---|---|---|---|---|---|---|---|---|---|
| ResNet18 | **67.21** | 66.90 | **68.26** | 67.26 | **69.23** | 68.57 | **70.04** | 68.74 | **68.84** | 67.79 | **67.29** | 66.51 |

*Table 16.* Ablation study 2: The contribution of constrained binary weights and scaling factors to improving environmental noise robustness of DBNNs on the Brain tumor dataset. The w/o indicates that the constraint on the scaling factors has been removed.

| Backbone | Dorefa+Our | Dorefa+Our+w/o | React+Our | React+Our+w/o | CycleBNN+Our | CycleBNN+Our+w/o |
|---|---|---|---|---|---|---|
| ResNet18 | **86.91** | 85.62 | **83.87** | 82.39 | **85.46** | 84.17 |

thereby evaluating the effectiveness of the proposed approach in defending adversarial attacks. Specifically, we use the LinfPGDAttack method from the library advertorch, with step size set to 7 and perturbation value set to $\epsilon = 1/255$. Then, we added such adversarial perturbations to the test set of the CIFAR-100 task and measured the CycleBNN with ResNet18 backbone and our proposed method. In Tab. 14, experimental results demonstrate that DBNN models, which have not been trained with adversarial samples, are highly susceptible to strong PGD attacks (Similar issues have been reported in full-precision models during both standard training and adversarial attack testing). The proposed method still has a certain percentage improvement despite the serious degradation of the performance of unconstrained DBNN models.

**Ablation Analysis for DBNNs:** We found that the constraints on binary weights and scaling factors are indispensable to improve the robustness of DBNNs.

