# OpenReview forum: "Can DBNNs Robust to Environmental Noise for Resource-constrained Scenarios?"
_ICML.cc/2025/Conference — ICML 2025 poster_

### Official Review · Reviewer_JCk1 · 2025-03-10

**Overall Recommendation:** 4

**Summary:**

This paper investigates the robustness of DBNNs under environmental noise in resource-constrained scenarios. The authors identify that the vulnerability of DBNNs stems from binary weights and scaling factors and propose an L1,∞-norm constraint to improve robustness. The proposed method introduces an auxiliary robustness loss function to balance classification and robustness objectives. Experiments on CIFAR-10, CIFAR-100, Brain Tumor MRI, and bio-electrical signal datasets show that the proposed method improves model accuracy and reduces accuracy degradation under noise compared to SOTA methods.

## Update after Rebuttal
The authors' rebuttal solved my concerns so I raised my score to 4.

**Claims And Evidence:**

The paper claims that L1,∞-norm constraints enhance DBNN robustness by improving the stability of binary weights and scaling factors under noise.  However, the evidence is limited to specific tasks and datasets, primarily focused on vision and medical diagnostics. Furthermore, while the paper reports lower computational overhead, it does not analyze real-time inference latency, which is important in resource-constrained scenarios.

**Essential References Not Discussed:**

No

**Experimental Designs Or Analyses:**

The experimental setup is well-organized and follows standard practices.

**Methods And Evaluation Criteria:**

The proposed method is clearly described. The authors provide a structured explanation of the L1,∞-norm constraint and its impact on DBNN robustness. The evaluation is based on widely used datasets and includes a reasonable selection of models (e.g., CNNs and vision transformers).

**Other Comments Or Suggestions:**

Figure 3 needs a clearer description of what w11 and w12 represent respectively

**Other Strengths And Weaknesses:**

The paper presents an interesting and potentially impactful method for improving DBNN robustness. However, the lack of generalization beyond vision and medical tasks reduces the scope of the contribution.

**Questions For Authors:**

1. How does the proposed method generalize to non-vision tasks such as NLP?
2. What is the real-time inference impact of the L1,∞-norm constraints, particularly in resource-limited environments?
3. Can you provide an ablation study to isolate the contribution of binary weights versus scaling factors in improving robustness?
4. How does the proposed method compare to more recent adaptive fault-tolerance methods beyond traditional redundancy-based techniques?

**Relation To Broader Scientific Literature:**

The paper builds on previous work in fault tolerance, particularly in binary neural networks and hardware-aware ML. It references relevant prior work and positions itself well within the context of DBNN optimization and robustness.

**Theoretical Claims:**

The paper provides a theoretical analysis of the robustness bounds derived from the L1,∞-norm constraints. The derivation appears sound, and the theoretical insights align with the empirical findings.

---

> ### Author Rebuttal · Authors · 2025-03-31
>
> Q1: How does the proposed method generalize to non-vision tasks such as NLP?
>
> A1: Thank you for your insightful question. From a theoretical perspective, extending to NLP tasks such as text classification or machine reading, it is necessary to first construct BERT models that binarize weights and activation, sort out their iterative and expansion forms, and the most critical step is to analyze the formalized upper bound of environmental noise perturbations for each layer. Because the structure of BERT is different from the DBNNs with ResNet backbone, it means that the Lipschitz constant of each part needs to be calculated carefully. This suggests that the upper bound of the binarized BERT model exhibits a slight deviation from Theorem 3. With a moderate adjustment, the theoretical analysis can be effectively applied to NLP tasks. On the other hand, the bio-electrical signal classification task and NLP task belong to the sequence modeling task. In particular, the data for the bio-electrical signal classification task are closely associated with time-dimensional information, and the proposed constraints have effectively validated the performance of the BNN model (e.g., A binary convolution layer as a replacement for the full-precision convolution layer).
>
> Q2: What is the real-time inference impact of the L1,∞-norm constraints, particularly in resource-limited environments?
>
> A2: Thank you for your nice question. In fact, the proposed constraints are incorporated into the training stage in the form of an objective function penalty term. This mean that the proposed method solely impacts the training cost, and thus no additional operations are introduced to the layers within the DBNNs. Consequently, it does not introduce extra time overhead during the model inference phase, thereby eliminating any inference concerns. Finally, DBNNs on the server side can achieve inference times at the second level.
>
> Q3: Can you provide an ablation study to isolate the contribution of binary weights versus scaling factors in improving robustness?
>
> A3: Thank you for your constructive suggestion. We found that the constraints on binary weights and scaling factors are indispensable to improve the robustness of DBNNs.
>
> Table 1. Ablation study 1: Binary weights V.S. Scaling factors to improving robustness on the CIRFA-100 dataset. The w/o indicates that the constraint on the scaling factors has been removed.
> | **Backbone** | **Dorefa+Our** | **Dorefa+Our+w/o** | **React+Our**  | **React+Our+w/o** | **ABC+Our**|**ABC+Our+w/o**| **IRNet+Our** | **IRNet+Our+w/o** | **Bireal+Our** |**Bireal+Our+w/o**|**CycleBNN+Our**|**CycleBNN+Our+w/o**|
> |:------------:|:--------------:|:------------------:|:--------------:|:-----------------:|:--------------:|:---------------:|:--------------:|:-----------------:|:--------------:|:------------------:|:----------------:|:--------------------:|
> | ResNet18| **67.21**| 66.90|**68.26**| 67.26|**69.23**|68.57|**70.04**|68.74|**68.84**|67.79|**67.29**|66.51|
>
> Table 2. Ablation study 2: Binary weights V.S. Scaling factors to improving robustness of DBNNs on the Brain tumor dataset.
> | **Backbone** | **Dorefa+Our** | **Dorefa+Our+w/o** | **React+Our**  | **React+Our+w/o** | **CycleBNN+Our** | **CycleBNN+Our+w/o** |
> |:------------:|:--------------:|:------------------:|:--------------:|:-----------------:|:----------------:|:--------------------:|
> | ResNet18|**86.91**|85.62| **83.87**| 82.39|**85.46**|84.17|
>
> Q4: How does the proposed method compare to more recent adaptive fault-tolerance methods beyond traditional redundancy-based techniques?
>
> A4: Thank you very much! The description of the manuscript may have given you some misunderstanding. We would like to clarify the distinction between the environmental noise robustness of DBNNs and adaptive error tolerance methods. Based on two recent works [1-2], adaptive fault tolerance methods focus on ensuring that hardware devices can continue performing tasks even after a fault occurs (i.e., The communication node is faulty). In contrast, the DBNN model exhibits approximately a 10-20% decrease in performance under environmental noise perturbations, rather than encountering fault. Therefore, we believe that it is not suitable to compare the adaptive fault-tolerance methods for hardware devices with the robustness improvement algorithm for DL models (DBNNs). Another reason is that the tasks targeted by these fault-tolerance algorithms differ significantly from the classification tasks of DBNNs, and their portability remains to be validated. We plan to investigate the effectiveness of adaptive fault-tolerance methods when the DBNN encounters fault.
>
> [1] Ada-FA: A Comprehensive Framework for Adaptive Fault Tolerance and Aging Mitigation in FPGAs. IEEE Internet Things J. 11(10): 17688-17699 (2024)
>
> [2] A Dynamic Adaptive Framework for Practical Byzantine Fault Tolerance Consensus Protocol in the Internet of Things. IEEE Trans. Computers 73(7): 1669-1682 (2024)

---

### Official Review · Reviewer_FmSJ · 2025-03-10

**Overall Recommendation:** 4

**Summary:**

This paper addresses the robustness of deep binary neural networks (DBNNs) under environmental noise perturbations in resource-constrained scenarios. The authors propose an $L_{1,\infty}$-norm constraint on binary weights and scaling factors to derive a tighter robustness upper bound compared to existing methods. Experiments on image classification (CIFAR, ImageNet) and medical tasks (bio-electrical signals, brain tumor MRI) demonstrate improved robustness with minimal computational overhead. While the work is well-motivated and technically sound, several aspects require strengthening for impact.

**Claims And Evidence:**

The claims in the paper are mostly well-supported. Specifically, the authors clearly demonstrate empirical gains compared to competitive baselines (ResNet 18/34, multiple BNN variants like BiRealNet, IR-Net, CycleBNN, etc.) on multiple image classification (CIFAR, ImageNet) and medical tasks (bio-electrical signals, brain tumor MRI) tasks. The theoretical claims regarding provided tightness bounds are sound and clearly explained.

**Essential References Not Discussed:**

The paper adequately discusses relevant literature.

**Experimental Designs Or Analyses:**

The experimental design is sound and thorough. Results across multiple DBNN-based models and five real-world datasets clearly illustrate the advantage of proposed method.
However, one minor limitation is that the paper does not provide the residual block analysis, or explicitly show how sensitive of the residual block is to environmental noise.

**Methods And Evaluation Criteria:**

The proposed robust training algorithm and the chosen evaluation criteria (five diverse image and medical tasks covering classification) are highly relevant and sensible for the problem studied.

**Other Comments Or Suggestions:**

1. Increase font size in Fig.4 for clearer visualization of feature maps.
2. Discuss potential risks of deploying noise-robust DBNNs in safety-critical medical applications (e.g., false negatives under extreme noise).

**Other Strengths And Weaknesses:**

Strengths
1. The paper identifies a critical gap in existing research: the lack of robustness analysis for binary neural networks (BNNs) under unpredictable environmental noise (e.g., patient movement artifacts). The proposed $L_{1,\infty}$-norm constraint offers a theoretically grounded solution, explicitly linking DBNN vulnerability to scaling factors and binary weights. The closed-form robustness bound (Theorem 4.2) is a key contribution, as it provides a quantifiable metric for evaluating DBNN robustness, surpassing the heuristic approaches of prior work (e.g., Shang et al., 2022).
2. The experiments span diverse domains (i.e., image classification, medical diagnostics) and architectures (ResNet 18/34, multiple BNN variants like BiRealNet, IR-Net, CycleBNN). Results show consistent robustness improvements (e.g., +5.4% on Brain Tumor MRI) while maintaining low computational overhead (16% faster training than LCR). The inclusion of real-world noise models (e.g., SNR=50% for bio-electrical signals) strengthens practical relevance.
3. The framework’s compatibility with various BNN backbones (e.g., ResNet34) and automated training pipeline (Fig.2) enhances applicability. The ablation study on constraint coefficients (also in the Appendix) and visualization of feature map perturbations (Fig.4) provide actionable insights for practitioners.
Weaknesses
1. This manuscript designs a large number of experiments to verify the validity of proposed method on different types of datasets. The reviewer wants to know how robust of proposed algorithm can be against adversarial attacks with latest BNNs (e.g., CycleBNN, Fontana et al., 2024).
2. The theoretical analysis assumes Lipschitz continuity of binarized activation but does not provide the residual block analysis. It is suggested that the author reflect this part of the analysis after the main theorem.

**Questions For Authors:**

1.The reviewer wants to know how robust of proposed algorithm can be against adversarial attacks with latest BNNs (e.g., CycleBNN, Fontana et al., 2024).
2. The theoretical analysis assumes Lipschitz continuity of binarized activation but does not provide the residual block analysis.

**Relation To Broader Scientific Literature:**

The paper appropriately relates its contributions to the broader literature on DBNN-based method under environmental noise perturbations in resource-constrained scenarios. It extends the current state-of-the-art by clearly showing the theoretical benefit of noise robustness, also alleviates the problem of introducing additional expensive costs with the SOTA approach.

**Theoretical Claims:**

The proofs and theoretical arguments appear technically correct and rigorous.

---

> ### Author Rebuttal · Authors · 2025-03-31
>
> Q1.The reviewer wants to know how robust of proposed algorithm can be against adversarial attacks with latest BNNs (e.g., CycleBNN, Fontana et al., 2024).
>
> A1. Thanks for the reviewer's nice suggestion. We opted for a standard training and testing phase to introduce PGD attacks, thereby evaluating the effectiveness of the proposed approach in defending adversarial attacks. This mode more closely with the hypothesis considered in our manuscript, which posits that noise is encountered exclusively during the inference phase. Specifically, we use the LinfPGDAttack method from the library advertorch, with step size set to 7 and perturbation value set to $\epsilon=1/255$. Then, we added such adversarial perturbations to the test set of the CIFAR-100 task and measured the CycleBNN with ResNet18 backbone and our proposed method. In below table, experimental results demonstrate that DBNN models, which have not been trained with adversarial samples, are highly susceptible to strong PGD attacks (Similar issues have been reported in full-precision models during both standard training and adversarial attack testing [A1]). The proposed method still has a certain percentage improvement despite the serious degradation of the performance of unconstrained DBNN models.
>
> Table 1. PGD performance comparison between CycleBNN and our on the CIRFA-100 dataset.
> | **Models**   | **Test PGD** |
> |:------------:|:------------:|
> | CycleBNN     | 17.78        |
> | CycleBNN+Our | **18.38** |
>
>
> [A1] Towards Deep Learning Models Resistant to Adversarial Attacks. ICLR (Poster) 2018.
>
> Q2. The theoretical analysis assumes Lipschitz continuity of binarized activation but does not provide the residual block analysis.
>
> A2. Thank you very much for your comment, we also think that the Lipschitz continuous analysis of residual structure is very important. Due to space constraints, the corollary and relevant proof of this part are located in the Appendix of the manuscript. Please refer to Corollary A.2 (Appendix) on page 12. For the camera version, we will move it into the main paper.
>
> Q3. Increase font size in Fig.4 for clearer visualization of feature maps.
>
> Q3. Please forgive our mistakes, we have set the font size to 24 to make the text description in Figure 4 clearer.
>
> Q4. Discuss potential risks of deploying noise-robust DBNNs in safety-critical medical applications (e.g., false negatives under extreme noise).
>
> A4. Thank you very much for your constructive suggestions. First, false negatives represent a significant challenge in the field of medical imaging. However, if imaging alone cannot provide a definitive determination, a biopsy can be conducted, and its subsequent pathological analysis serves as the gold standard for diagnosis. In practical MRI, extreme noise perturbations caused by patient movement can occur. In such cases, experienced senior physicians on the medical team will implement appropriate corrective measures. This indicates that our proposed method is unlikely to encounter such issues under normal environmental noise conditions.

---

> > ### Comment · Reviewer_FmSJ · 2025-04-05
> >
> > Thank you for your detailed feedback. I have read your response. You have provided a good feedback which addresses most concerns. I have raised the score to 4, hope that authors include these proposed revisions in your final version.

---

> > > ### Author Response · Authors · 2025-04-05
> > >
> > > Dear Reviewer FmSJ,
> > >
> > > Thank you very much! We are pleased to address your main concerns and will incorporate the aforementioned four revisions into the final version.
> > >
> > > Best regards,
> > > All authors

---

### Official Review · Reviewer_1Wt1 · 2025-03-14

**Overall Recommendation:** 3

**Summary:**

The paper investigates whether Deep Binary Neural Networks (DBNNs) can be robust to environmental noise, particularly in resource-constrained scenarios such as bio-electrical signal classification and medical imaging. The authors identify that DBNNs' robustness vulnerabilities stem from binary weights and scaling factors. To address this, they propose a L1,∞-norm constraint for binary weights and scaling factors, which they claim provides a tighter upper bound on noise perturbations compared to state-of-the-art (SOTA) methods.

Their approach involves:
1. Theoretical Analysis: They derive a formal noise perturbation upper bound for DBNNs using L1,∞-norm constraints.
2. Robust Training Framework: The proposed L1,∞-norm constraint is incorporated into the training process to enhance DBNN robustness.
3. Experimental Validation: Their method is tested on five classification datasets, including CIFAR-10, CIFAR-100, Brain Tumor MRI, and bio-electrical signal datasets. Results show improved robustness, with up to 4.8% and 5.4% improvements on CIFAR-100 and Brain Tumor MRI, respectively.
4. Computational Efficiency: Their method reduces additional training overhead compared to previous methods.

The study concludes that L1,∞-norm constraints effectively mitigate the impact of environmental noise on DBNNs while maintaining efficiency, making it suitable for edge devices in safety-critical tasks.

**Claims And Evidence:**

1. Claim: DBNNs' robustness vulnerability comes from binary weights and scaling factors.
Evidence: The authors provide a theoretical analysis showing that binary weights and scaling factors contribute to noise sensitivity. They derive an upper bound on noise perturbations using L1,∞-norm constraints, which is compared to prior approaches.

2. Claim: The proposed L1,∞-norm constraint provides a tighter upper bound than existing methods.
Evidence: Theoretical derivations show that the L1,∞-norm constraint offers a more restrictive bound on noise effects compared to spectral norm-based constraints. This is further supported by quantitative comparisons.

3. Claim: The proposed method improves robustness across multiple DBNN architectures.
Evidence: The authors conduct experiments on five classification datasets (CIFAR-10, CIFAR-100, Brain Tumor MRI, bio-electrical signals). The results show robustness improvements (up to 4.8% on CIFAR-100 and 5.4% on Brain Tumor MRI), validating their claim.

4. Claim: The method reduces additional computational overhead compared to previous robustness approaches.
Evidence: They compare training/testing time with the LCR (Shang et al., 2022) method and show a 16% reduction in training time while maintaining performance.

Problematic Claims:
*** Claim: The method is broadly applicable to real-world safety-critical tasks.
Issue: The experiments are conducted only on standardized datasets, without real-world deployment on actual medical or edge devices. Practical applicability remains untested. Also, the CIFAR datasets are considered too small scaled for now-a-day studies.

*** Claim: The proposed method is universally effective across DBNN architectures.
Issue: The study focuses mainly on ResNet-based DBNNs. It is unclear how well it generalizes to non-ResNet architectures or tasks beyond classification. for example transformer architectures.

**Essential References Not Discussed:**

I'm not familiar with this field

**Experimental Designs Or Analyses:**

The experimental design is generally well-structured for evaluating the robustness of DBNNs to environmental noise, using five datasets (CIFAR-10, CIFAR-100, Brain Tumor MRI, bio-electrical signals, and ImageNet) and measuring classification accuracy under noise perturbations. However, the study lacks real-world deployment tests on edge devices, making it unclear how well the method generalizes to practical applications. The noise perturbation strategy is reasonable but does not appear to be based on real-world noise distributions, which could affect its applicability. While the comparison against four BNN-based models and the LCR method (Shang et al., 2022) is fair, the study does not include adversarial robustness methods, which would provide a more complete assessment. Additionally, while computational efficiency is analyzed in terms of training time, there is no analysis of inference speed and memory usage on actual resource-limited devices, which is crucial for assessing its practical feasibility. Overall, the experiments demonstrate improvements in robustness but leave important questions about real-world applicability unanswered.

**Methods And Evaluation Criteria:**

Yes, the proposed methods and evaluation criteria make sense for the problem. The L1,∞-norm constraint is a reasonable approach to enhance robustness in Deep Binary Neural Networks (DBNNs), and the authors justify its effectiveness through theoretical derivations and empirical validation.

For evaluation, the authors use five benchmark datasets (CIFAR-10, CIFAR-100, Brain Tumor MRI, bio-electrical signals, and ImageNet), which are commonly used in robustness studies, but are fairly small scale. They introduce environmental noise perturbations and measure test accuracy under noise, which is an appropriate metric for assessing robustness.

**Other Comments Or Suggestions:**

N/A

**Other Strengths And Weaknesses:**

The paper presents an original approach by adapting Lipschitz-based robustness constraints to DBNNs, offering a novel L1,∞-norm constraint that improves robustness with lower computational cost, which is a meaningful contribution to lightweight model research. It effectively highlights a practical issue—environmental noise affecting DBNNs in resource-constrained scenarios—that has been underexplored compared to adversarial robustness.

However, its significance is somewhat limited by the lack of real-world deployment or validation on actual edge devices, making it unclear how well the method translates to practical applications. The clarity of theoretical explanations and experimental results is generally strong, but some assumptions, such as the applicability of the bound across architectures, lack thorough empirical verification. While the work is an important step toward improving DBNN robustness, further validation in real-world conditions would enhance its impact.

**Questions For Authors:**

N/A

**Relation To Broader Scientific Literature:**

The paper builds on prior work in BNNs and robustness methods by addressing DBNNs' vulnerability to environmental noise, extending Lipschitz-based constraints (Gouk et al., 2021; Miyato et al., 2018) with an L1,∞-norm constraint that is more efficient than the LCR method (Shang et al., 2022). It improves robustness for resource-constrained applications like medical signal processing, complementing adversarial robustness studies (Gowal et al., 2018; Balunovic & Vechev, 2020) by focusing on random noise instead of attacks.

**Theoretical Claims:**

The proofs seem mathematically valid and logically structured. However, the assumptions about scaling factors and generalization to different architectures require additional empirical verification. A formal empirical validation comparing actual perturbation bounds across architectures would strengthen confidence in the claims.

---

> ### Author Rebuttal · Authors · 2025-03-31
>
> Many thanks to the reviewers for their positive comments and constructive comments.
>
> Q1. However, its significance is somewhat limited by the lack of real-world deployment or validation on actual edge devices.
>
> A1. We apologize for the fact that the inference experiments on actual edge devices.  However, the PyTorch framework can integrate ONNX third-party libraries to support DBNN model transformation and facilitate deployment on the simulation platform of edge devices (i.e., Raspberry Pi Debian 12). Specifically, the CycleBNN method with a ResNet18 backbone, after being trained with our $L_{1,\infty}$-norm constraint loss, resulted in a checkpoint file size of 45.34MB. This is significantly smaller than the 8GB storage capacity of the Raspberry Pi 4B+ device. The inference of a single image under environmental noise takes only 1.6 seconds, which is deemed acceptable given that brain tumor-related tasks typically do not demand real-time processing capabilities. In particular, the performance of the Raspberry PI simulation environment provided by Docker is lower than that of the actual device. This indicates substantial potential for deploying our proposed method on the actual edge device.
>
> Q2. The clarity of theoretical explanations and experimental results is generally strong, but some assumptions, such as the applicability of the bound across architectures.
>
> A2. Thank you for the reviewer's valuable suggestions. We agree that the applicability of the bound across architectures plays an important role in ensuring the theoretical robustness analysis of DBNNs under environmental noise perturbations. We clarify that the DBNN-based model discussed primarily focuses on the binarization of weights and activations of CNN-based models. However, the multi-head self-attention mechanism in transformer-based models differs significantly from the convolution operations in CNNs, and ResNet-based backbones also lack a positional embedding layer. Thus, potential issues for applications of bound across architectures are more likely to stem from substantial architectural differences rather than from the assumptions of theoretical analysis. Nevertheless, we will conduct a rigorous analysis of the upper bound of environmental noise perturbations in both the position embedding layer and the multi-head self-attention layer after binarization, thereby extending the existing conclusions to the Transformer model.

---

### Official Review · Reviewer_PX4V · 2025-03-17

**Overall Recommendation:** 3

**Summary:**

In this work, the authors investigate the robustness of deep binary neural networks (DBNNs) under environmental noise perturbations in resource-constrained scenarios. Specificity, the authors propose an $L_{1,\infty}$-norm constraint on objective function for binary weights to derive a tighter robustness upper bound and a low computational overhead training algorithm compared to existing methods. Then, the authors conduct extensive experiments on three benchmark image datasets (i.e., CIFAR-10, 100 & ImageNet) as well as medical datasets (i.e., bio-electrical signals, brain tumor MRI), validating the proposed algorithm by enhancing the robustness of major BNN algorithms.

## Rebuttal summary

This paper studies an interesting problem about the robustness of deep binary neural networks under environmental noise perturbations in resource-constrained scenarios. My initial concerns have been resolved, so I keep a positive score.

**Claims And Evidence:**

In this work, the claims are well-supported. Firstly, the theoretical perspective establishes a tighter upper bound on environmental noise robustness compared to prior studies and further elucidates the quantitative relationships, thereby offering a clear explanation of the robustness in DBNNs-based model during inference. Finally, the authors conduct extensive experiments on three benchmark image datasets (i.e., CIFAR-10, 100 & ImageNet) as well as medical datasets (i.e., bio-electrical signals, brain tumor MRI), validating the proposed algorithm by enhancing the robustness of major BNN algorithms.

**Essential References Not Discussed:**

This manuscript discusses the relevant literature.

**Experimental Designs Or Analyses:**

The experimental setup of this manuscript is accurate and comprehensive. The experimental results are well verified on three image classification datasets. In addition, the advantages of the algorithm in medical time series and image tasks are fully visualized.
It would be better if the authors could provide more analysis of the computational overhead.

**Methods And Evaluation Criteria:**

The proposed method targets the interference of environmental noise during the inference stage. Specifically, the author uses test accuracy in a noisy environment as the evaluation metric, which is highly relevant to the research areas.

**Other Comments Or Suggestions:**

1. Authors should give priority to the visual results of medical image classification in the main text. After all, the results on real tasks are better than tables to visualize the advantages of the proposed approach.
2. It would be better if the authors could provide more BNN-based algorithm overhead under environment noise on the medical dataset.

**Other Strengths And Weaknesses:**

Strengths
1. The background of the issue examined in this manuscript is both critical and intriguing, particularly considering the paucity of theoretical studies that have explored the robustness of DBNN in environmental noise scenarios.
2. In this paper, a low-cost and robust training algorithm is developed from the perspective of dual norm constraints, and a more compact and noise-robust upper bound is presented.
3. The visualization results for both medical image and standard image classification tasks demonstrate the significant effectiveness of the proposed algorithm. Specifically, it elucidates the extent to which ambient noise perturbation interferes with the binarized convolution component of the model.
Weaknesses
1. This manuscript conducts several comprehensive experiments to verify the advantage of the proposed method on different types of datasets. It seems that model overhead analysis is only available on the ImageNet dataset. It would be nice to analyze the overhead of more BNN algorithms under noise perturbations on other datasets.

**Questions For Authors:**

1. The reviewer wants to know more about the cost of the BNN algorithm under environmental noise perturbations.

**Relation To Broader Scientific Literature:**

The paper effectively situates its contributions within the broader scientific literature of DBNN-based methods, specifically addressing environmental noise perturbations in resource-constrained scenarios. It mainly explains the robustness of DBNNs in specific scenarios, and also relieves the problem of high robust training cost faced by existing methods.

**Theoretical Claims:**

The theoretical analysis of this manuscript is accurate and rigorous.

---

> ### Author Rebuttal · Authors · 2025-03-31
>
> Q1: Authors should give priority to the visual results of medical image classification in the main text.
>
> A1: We extend our gratitude to the reviewer for providing this constructive suggestion. We fully concur with the suggestion to include visual results in the main text to effectively demonstrate the efficacy of the proposed method. The visualized results of brain tumors have been relocated to the experiment section.
>
> Q2: The reviewer wants to know more about the cost of the BNN algorithm under environmental noise perturbations.
>
> A2: Thank you very much for the valuable comment. We concur that, in addition to comparing the computational overhead with the LCR method (SOTA), which investigates BNN robustness, it is also essential to explore how the proposed $L_{1,\infty}$-norm constraints loss function influence the computational overhead of various popular BNN algorithms. To expedite the acquisition of experimental results, we measured the actual computation time on an A800 GPU.
>
> Specifically, we have evaluated the training cost of a series of BNN-based methods on CIFAR-100 and Brain Tumor datasets. We have reported the average training time of each epoch and total test time. According to the results presented in the main paper and two table below, it can be observed that an appropriately designed penalty term not only enhances model robustness but also significantly improves training efficiency. In addition, the overhead in the test phase also encompasses the time required for adding environmental noise. The detailed computational overhead are presented below:
>
> Table 1. Computational cost comparison between popular BNN-based methods and our on the CIFAR-100 dataset.
> | **Methods (ResNet18 backbone)** | **Training Time/Epochs (s)** | **Total Test Time (s)** |
> |:-------------------------------:|:----------------------------:|:-----------------------:|
> | CycleBNN                        | 24                           | 43                      |
> | CycleBNN+Our                    | **21**                    | **42**             |
> | IRNet                           | 13                           | 32                      |
> | IRNet+Our                       | **12**                  | **31**            |
> | Dorefa                          | 11                           | 33                      |
> | Dorefa+Our                      | 11                           | 33                      |
> | React                           | 14                           | 33                      |
> | React+Our                       | **13**                  | 33                      |
> | Bireal                          | 13                           | 30                      |
> | Bireal+Our                      | **12**                  | 30                      |
> | ABCNet                          | 13                           | 41                      |
> | ABCNet+Our                      | **12**                  | 41                      |
>
> Table 2. Computational cost comparison between BNNs and our on the Brain tumor dataset.
> | **Methods (ResNet18 backbone)** | **Training Time/Epochs (s)** | **Total Test Time (s)** |
> |:-------------------------------:|:----------------------------:|:-----------------------:|
> | CycleBNN                        | 13                           | 8                       |
> | CycleBNN+Our                    | **11**                    | 8                       |
> | Dorefa                          | 11                           | 3                       |
> | Dorefa+Our                      | **10**                 | 3                       |
> | React                           | 6                            | 4                       |
> | React+Our                       | **5**                   | 4                       |

---

### Decision · Program_Chairs · 2025-05-01

**Decision:**

Accept (poster)

**Comment:**

The paper investiages the robustnes of deep binary neural networks (DBNNs) for compute resource constrained settings under noise perturbations. The paper proposes a method, inspired by theoretical analysis, that improves robustness.

The paper's four reviewers all recommend acceptance after the rebuttal.

Strength:
* Reviewer PX4V find the theoretical analysis and experiments sound an rigorous and the algorithm to be effective.
* Reviewer 1Wt1 finds two claims problematic, the claim of broad applicability and the claim of reducing computational overhead. Specifically, the method is only investigated for ResNet type architectures. The authors remark that deep binary neural networks are primarily CNN based networks as investigated in the paper.
* Reviewer FmSJ also finds the theoretical claims and analysis to be well-supported and the advantage of the method to be clearly demonstrated.

The paper provides a sound and well-verified contribution to the robustness of deep binary neural networks. I therefore recommend the paper for publication.